# Leave-One-Trial-Out, LOTO, a general approach to link single-trial parameters of cognitive models to neural data

Sebastian Gluth[1]*, Nachshon Meiran[2,3]

[1]Department of Psychology, University of Basel, Basel, Switzerland; [2]Department of Psychology, Ben-Gurion University of the Negev, Beer-Sheva, Israel; [3]Zlotowski Center for Neuroscience, Ben-Gurion University of the Negev, Beer-Sheva, Israel

**Abstract** A key goal of model-based cognitive neuroscience is to estimate the trial-by-trial fluctuations of cognitive model parameters in order to link these fluctuations to brain signals. However, previously developed methods are limited by being difficult to implement, time-consuming, or model-specific. Here, we propose an easy, efficient and general approach to estimating trial-wise changes in parameters: Leave-One-Trial-Out (LOTO). The rationale behind LOTO is that the difference between parameter estimates for the complete dataset and for the dataset with one omitted trial reflects the parameter value in the omitted trial. We show that LOTO is superior to estimating parameter values from single trials and compare it to previously proposed approaches. Furthermore, the method makes it possible to distinguish true variability in a parameter from noise and from other sources of variability. In our view, the practicability and generality of LOTO will advance research on tracking fluctuations in latent cognitive variables and linking them to neural data.

DOI: https://doi.org/10.7554/eLife.42607.001

*For correspondence:
sebastian.gluth@unibas.ch

**Competing interests:** The authors declare that no competing interests exist.

## Introduction

Model-based cognitive neuroscience attempts to link mathematical models of cognitive processes to neural data in order to advance our knowledge of the mind and brain (*Forstmann et al., 2011*). A particularly promising but challenging approach in this regard is to derive trial-specific values for parameters of cognitive models and to relate these values to trial-specific brain data, which offers insights into cognitive and neural principles at a highly detailed level of analysis (*Gluth and Rieskamp, 2017*; *van Maanen et al., 2011*; *Turner et al., 2017*; *Wiecki et al., 2013*). In the present work, we introduce a novel technique for capturing trial-specific values of model parameters that is efficient, widely applicable and easy to apply. We briefly summarize related work and methods before turning to our current proposal.

Primarily, the difficulty in linking momentary fluctuations in cognitive states to neural measures lies in specifying the variability in the cognitive model. Often, cognitive models specify the distribution from which this variability is assumed to originate (e.g., a normal distribution), but remain silent about its direction and extent in single trials. Previous attempts to capture parameter variability have often been specific to a single model. For example, *van Maanen et al. (2011)* derived maximum likelihood estimates for two single-trial parameters of the Linear Ballistic Accumulator (LBA) model (*Brown and Heathcote, 2008*). Because the LBA model is of special interest for the present paper, we describe it in some detail here. The LBA belongs to the class of sequential sampling models of decision making, which assume that decisions arise from an evidence-accumulation process that continues until a threshold has been reached, indicating that the required amount of evidence for a decision has been gathered. Three critical parameters of the LBA are the rate of evidence

accumulation (drift rate), the amount of evidence required to reach a decision (decision threshold), and the point from which the accumulation begins (start point). Importantly, sequential sampling models such as LBA predict accuracy rates and the shape of response times (RT) distributions conjointly. *Wiecki et al. (2013)* introduced a hierarchical Bayesian modeling tool for another sequential sampling model, the Diffusion Decision Model (DDM) (*Ratcliff, 1978*). Their tool allows the regresssion of single-trial parameters of the DDM onto trial-by-trial neural measures such as electroencephalography (EEG) or event-related functional magnetic resonance imaging (fMRI) recordings (similar regression-based approaches were proposed by *Hawkins et al. (2017)* and by *Nunez et al. (2017)*). Turner and colleagues proposed a related method, whose rationale is to model behavioral and neuroimaging data jointly under a common (hierarchical Bayesian) umbrella (*Palestro et al., 2018*; *Turner et al., 2013*; *Turner et al., 2015*). Although the approach was again tested within the DDM framework, it may be generalized to any other cognitive model. As we have argued before, however, the complexity of this approach might discourage many cognitive neuroscientists from applying it (*Gluth and Rieskamp, 2017*). Another, but less general, approach of modeling behavioral and brain data jointly was proposed by *van Ravenzwaaij et al. (2017)*.

We recently proposed a comparatively simple and very general approach to capture trial-by-trial variability in cognitive model parameters (*Gluth and Rieskamp, 2017*). Again, the approach is based on Bayesian principles: it specifies the posterior probability of a parameter value in a specific trial, using the 'average' parameter value across all trials and the behavior in the specific trial as the prior and likelihood, respectively. Thus, the method basically answers the question of how the difference between a person's behavior in specific trials vs. that in all trials can be mapped (in a Bayesian optimal way) onto changes in a specific parameter of interest. For example, a surprisingly fast decision made by an otherwise very cautious decision maker might be attributed to a reduced decision threshold (in the context of sequential sampling models such as LBA or DDM), which then might be linked to altered neural activation in a specific region of the brain such as the pre-supplementary motor area (*Gluth et al., 2012*; *van Maanen et al., 2011*). While this approach is both general and comparatively simple, it can require high amounts of computation time, in particular when one seeks to estimate variability in more than one parameter simultaneously.

In the current work, we propose a novel approach to capturing trial-by-trial variability in cognitive model parameters that tries to overcome the shortcomings of specificity, complexity, and inefficiency: Leave-One-Trial-Out (LOTO). Briefly, the idea of LOTO is that the difference in the parameter estimates that are based on all trials vs. those that are based on all trials except a single trial provides a reflection of the 'true' parameter value in the left-out trial. The rest of the article is structured as follows: first, we introduce the rationale behind LOTO and exemplify this rationale with a simple toy model. Second, we show the circumstances under which LOTO can be expected to capture trial-by-trial variability appropriately, and explain how LOTO's performance can be improved by adapting its application, the underlying cognitive model or the experimental design. Third, we illustrate how one can test the assumption of the presence of systematic parameter variability. Fourth, we compare LOTO to previously proposed methods. Fifth, we present a simulation-based power analysis that aims to test whether LOTO can be expected to identify the neural correlates of trial-by-trial fluctuations of model parameters with sufficient power and specificity. Finally, we provide an example analysis, in which we apply LOTO to link variability in a cognitive process (encoding of episodic memory) to brain signals (fMRI activation in the hippocampus). Although we believe that it is essential for the potential user to read the entire article (in order to understand when and why LOTO can be expected to provide acceptable results), we note that some parts of the article are rather technical and may be difficult to understand. Therefore, our suggestion for readers with limited statistical and mathematical knowledge is to read 'Results: II. The LOTO principle', and 'Results: XI.: An example of using LOTO for model-based fMRI', which exemplifies the application of LOTO, and then to proceed directly to the discussion, in which we provide a 'LOTO recipe' that provides a step-by-step summary of how LOTO should be applied. The recipe then refers the reader back to relevant specific sections of the article.

## Results

### I. The LOTO principle

The rationale behind LOTO is simple and intuitive and can be summarized as follows:

1. When fitting a (cognitive) model to data, all $n$ trials contribute to the model's goodness-of-fit statistic (e.g., log-likelihood) and thus to the estimation of parameter values.
2. A single trial, $t$, shifts the parameter estimation to the direction that accords with the behavior in trial $t$.
3. Therefore, if trial $t$ is taken out and the model is fitted again to the $n–1$ dataset, the new parameter estimate will shift (slightly) to the opposite direction.
4. Therefore, the (true and unknown) parameter value in $t$ is reflected in the difference between the parameter estimate of all trials and the parameter estimate of all trials except $t$.

Accordingly, the application of LOTO works as follows:

1. Estimate all parameters of your model by fitting the model to the entire dataset (e.g., all $n$ trials of a subject).
2. Re-estimate the parameter(s) of interest by fitting the model to the dataset without trial $t$; keep all other parameters fixed to their estimates derived in step 1.
3. Take the difference between the parameter estimate(s) of step 1 and step 2, which will provide a reflection of the parameter value in $t$ (i.e., the difference should be correlated positively with the true parameter value).
4. Repeat the Steps 2 and 3 for all $n$ trials.
5. You may then link the obtained vector of trial-by-trial parameter values to any external measure that was recorded at the same time and that has an appropriate temporal resolution (e. g., event-related fMRI data, EEG data, single-unit recordings, eye-tracking data, skin conductance responses; see also *Gluth and Rieskamp, 2017*).

Note that instead of taking the difference between the estimates for $n$ and $n–1$ trials in step 3, one could simply multiply the $n$-1 trials estimates by $−1$ to obtain LOTO estimates that are positively correlated with the true parameter values. However, we prefer taking the difference, as this nicely separates above-average from below-average parameter values, which will be positive and negative, respectively. In general, it should be obvious that LOTO does not provide single-trial parameter estimates in an absolute sense. Rather, the method provides information about the direction and (relative) amount of deviation of single-trial parameter values from the average. Note that, in most cases, cognitive neuroscientists will be interested in the correlation between trial-by-trial parameter values and neural data, for which absolute values do not matter (*Gluth and Rieskamp, 2017*). In the rest of the article, we sometimes refer to 'LOTO estimates of a parameter', but it should be clear that this refers to estimates in a relative sense only.

### II. A 'toy' model example

In the following, we illustrate the LOTO principle using the binomial distribution as a 'toy' model example. In general, the binomial distributions specifies the probability of observing $k$ successes out of $n$ trials given probability $\theta$, which specifies the probability of a success in each trial. Let us assume that we seek to find the estimate of $\theta$ that provides the best account of the observation of $k$ successes in $n$ trials. The Maximum Likelihood Estimate (MLE) of this is simply (e.g., *Farrell and Lewandowsky, 2018*):

$$\mathrm{MLE}(\theta) = \frac{k}{n} \tag{1}$$

Let us now assume that we believe that $\theta$ is not stable but may vary from trial to trial, and that our goal is to estimate this trial-by-trial variability using LOTO. (Note that for this and the following sections, we simply assume that such trial-by-trial variability exists; we address the issue of testing for the presence of trial-by-trial variability in 'Results: Section V'. Also note that for this 'toy' model example, LOTO cannot provide more information about parameter $\theta$ than what is already given by the observations $k$; the goal of this section is to exemplify the workflow of LOTO rather than its capabilities.) As explained above, LOTO's estimate for trial $t$ is the difference between the $n$ and the $n–1$ estimates of $\theta$. For the binomial model, we thus obtain:

$$\mathrm{LOTO}_t = \mathrm{MLE}(\theta) - \mathrm{MLE}(\theta)_{\neg t} = \frac{k}{n} - \frac{k - k_t}{n - 1}, \tag{2}$$

where $k_t$ indicates whether a success was observed in trial $t$ ($k_t = 1$) or not ($k_t = 0$). The critical question is whether we can expect the LOTO approach to capture changes in observations meaningfully. To answer this question, we take the derivative of *Equation 2* with respect to $k_t$ to see how LOTO's estimate changes as a function of $k_t$:

$$\frac{\partial \mathrm{LOTO}_t}{\partial k_t} = \frac{1}{n - 1}. \tag{3}$$

*Equation 3* implies that changes in single-trial observations map linearly and positively onto LOTO's estimates, which is what we desired. To give an example, let us assume $k = 5$, $n = 10$, $k_{t=1} = 1$, and $k_{t=2} = 0$. Then LOTO's estimates for $t = 1$ and $t = 2$ are 1/18 and −1/18, respectively, with the difference being $1/(n–1) = 1/9$. *Equation 3* also reveals a general property of LOTO, namely that the change in LOTO is inversely related to the number of trials: naturally, the influence of a single trial on the estimation of a parameter decreases as the number of trials increases. We will further elaborate on this issue in 'Results: Section VII'. Also, we see that $\mathrm{LOTO}_t$ is not an estimate of $\theta$ for trial $t$ in an absolute sense (e.g., a $\theta$ of −1/18 is impossible), but that positive and negative $\mathrm{LOTO}_t$ values indicate above- and below-average single-trial values, respectively.

So far, we have only established a sensible relationship between LOTO's estimate and the observation $k_t$, but ultimately we are interested in whether LOTO is also related to the underlying trial-wise (data-generating) parameter $\theta_t$. This depends on how a change in $\theta_t$ is mapped onto a change in $k_t$, which is given by the Fisher information, the variance of the first derivative of the log-likelihood function with respect to $\theta_t$. For the binomial distribution with $n = 1$ (i.e., the Bernoulli distribution), the Fisher information is (e.g., *Ly et al., 2017*):

$$I(\theta) = \frac{1}{\theta * (1 - \theta)}. \tag{4}$$

As can be seen in the left panel of *Figure 1A*, the Fisher information of the Bernoulli distribution indicates that observations are more informative about the underlying parameter at the extremes. Accordingly, LOTO's performance (i.e., the correlation between the true $\theta_t$ and LOTO's estimates of it) will also depend on the range of $\theta_t$. To illustrate this, we ran simulations by drawing 300 values of $\theta_t$ from a uniform distribution with a range of .2 (e.g., between .4 and .6) and repeated this for every range from [0, .2] to [.8, 1] in steps of .01 (detailed specifications of all simulations are provided in 'Materials and methods'). LOTO was then used to recover $\theta_t$. The middle panel of *Figure 1A* depicts the average correlation between $\theta_t$ and its LOTO estimate as a function of the average $\theta$. As predicted by the Fisher information, the correlation is higher when $\theta$ is close to 0 or close to 1. Even though we do not further discuss the Fisher information in the context of the more complex models in the following sections, the principles of Fisher information also apply there (including the fact that more extreme parameter values are most informative).

Notably, the (average) correlations between $\theta_t$ and LOTO's estimate are quite low (i.e., < .2). The reason for this can be seen in the right panel of *Figure 1A*, which shows the correlation for an example of 300 trials in the range [.4, .6]: the estimate of LOTO can only take two values, depending in whether a success was observed in $t$ or not. Fortunately, the performance can be improved by assuming that $\theta$ does not change from trial to trial but remains constant over a certain number of trials, which we denote $m$. The larger $m$, the more different values LOTO can take (and at the same time, our confidence in the estimate of $\theta$ increases). LOTO's estimate, its derivative with respect to $m$, and the Fisher information for the Bernoulli distribution generalized to $m \geq 1$ (i.e., the binomial distribution) are:

$$\mathrm{LOTO}_m = \mathrm{MLE}(\theta) - \mathrm{MLE}(\theta)_{\neg m} = \frac{k}{n} - \frac{k - k_m}{n - m}, \tag{5a}$$

$$\frac{\partial \mathrm{LOTO}_m}{\partial k_m} = \frac{1}{n - m}, \text{ and} \tag{5b}$$

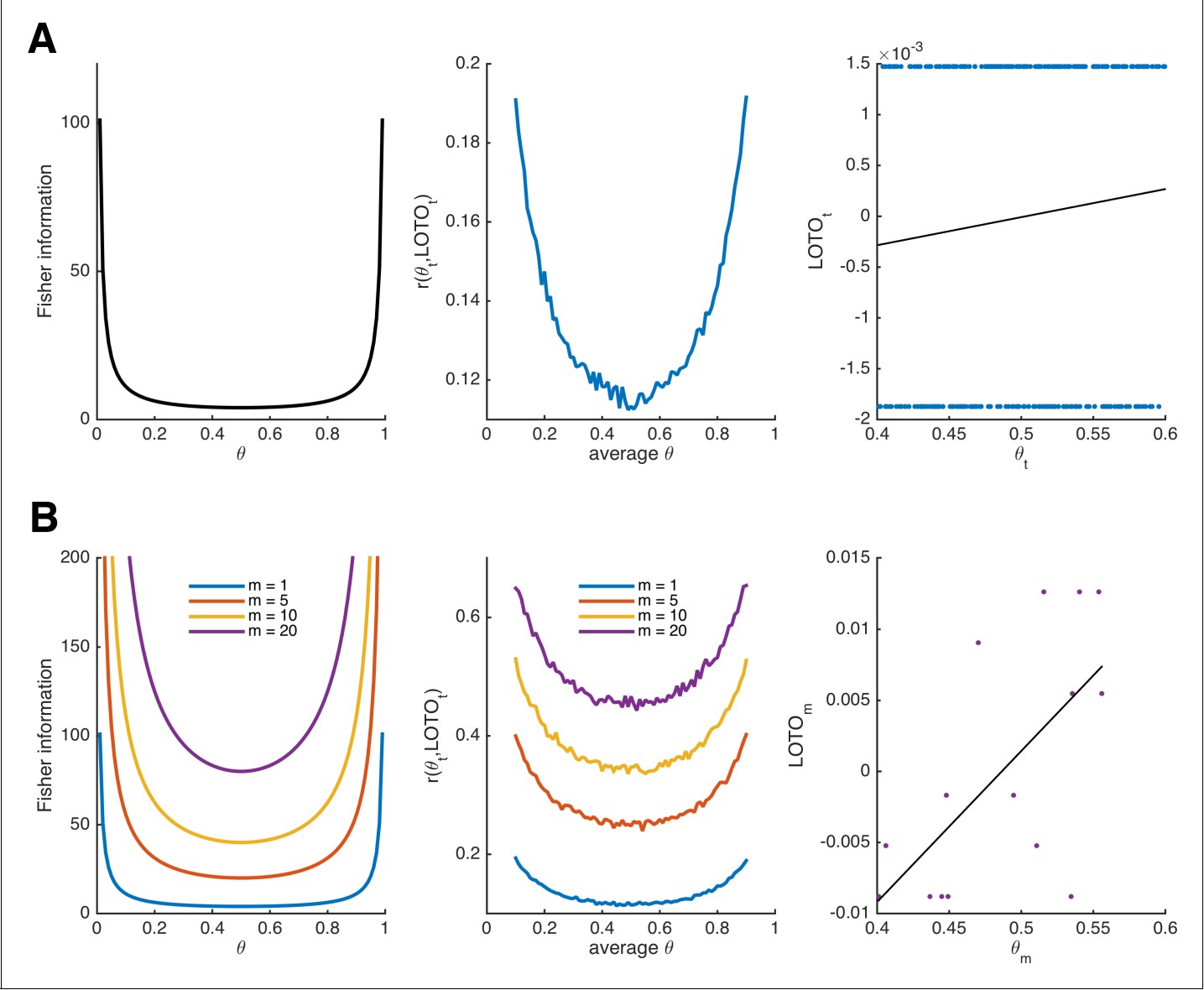

**Figure 1.** Fisher information and performance of LOTO for the binomial distribution. (A) Fisher information (left panel), average correlation between and LOTO's estimate (middle panel), and example correlation for values of $\theta_t$ in the range [.4,.6] (right panel) for the Bernoulli distribution. (B) The same as in panel A but generalized to the binomial distribution, that is, assuming that $\theta$ is stable over $m$ trials, for different levels of $m$ (the example correlation in the right panel uses $m = 20$).

DOI: https://doi.org/10.7554/eLife.42607.002

$$I(\theta) = \frac{m}{\theta * (1 - \theta)}, \text{ respectively.} \tag{5c}$$

*Figure 1B* illustrates the increases in Fisher information and LOTO's performance as $m$ increases. This is true, even though we kept the number of draws to 300 (implying that when $m = 20$, there are only $n/m = 300/20 = 15$ values estimated by LOTO; see *Figure 1B*, right panel). Thus, it might be more promising to track fluctuations over multiple trials (e.g., short blocks or sessions) than to try to capture parameter values in each and every trial. For example, *Gluth et al. (2015)* investigated memory-based decisions and inferred whether an item had been remembered on the basis of three trials that included the item as a choice option (see also 'Results: Section XI', in which we use LOTO to reproduce this analysis).

We see that LOTO can be applied to the Bernoulli and binomial distributions, but it should be (again) noted that LOTO is not particularly helpful in these cases. This is because LOTO will not provide us with any novel information over and above what is already known from the observations. In other words, LOTO's estimates are perfectly correlated with $k_t$ or $k_m$, which is mirrored in the derivatives (*Equations 3 and 5b*) which depend only on constants $n$ or $m$. In principle, there are two ways to make LOTO (and any other method for tracking trial-by-trial variability in parameters) more useful. First, one could introduce variations in the task, with these variations being directly linked to the predictions of the cognitive model. Second, one could apply cognitive models that predict a 'richer' set of data (e.g., sequential sampling models that predict accuracy rates and RT distributions conjointly; see above). In the next section, we first turn to introducing variations in the task.

## III. Task variations and comparison with 'single-trial fitting'

We demonstrate the importance of using variations in the experimental design that are systematically linked to the prediction of the cognitive model within the context of studying intertemporal choices. Intertemporal choices are decisions between a smaller reward available now or in near future vs. a larger reward available only in the more distant future (an example trial of a typical intertemporal choice task could be: *'Do you want to receive $20 now or $50 in 20 days?'*). Traditionally, these decisions have been modeled using temporal discounting models that describe the decrease in utility $u_i$ of an option $i$ as a function of the delay $d_i$ of its pay-out (*Frederick et al., 2002*). We apply the hyperbolic discounting (HD) model (*Mazur, 1987*), according to which:

$$u_i = \frac{x_i}{1 + \kappa * d_i},$$  (6)

where $x_i$ refers to the amount of reward and $\kappa$ is the discount rate (i.e., a free parameter modeling the degree to which utilities are lowered by delays; $0 \leq \kappa \leq 1$). To derive choice probabilities (that can then be subjected to MLE for the estimation of $\kappa$), the HD model is usually combined with the logistic (or soft-max) choice function (e.g., *Gluth et al., 2017*; *Peters et al., 2012*), according to which the probability $p_i$ of choosing option $i$ from a set of options including $i$ and $j$ is:

$$p(i|\{i,j\}) = p_i = \frac{1}{1 + \exp\left(-\beta * [u_i - u_j]\right)},$$  (7)

where $\beta$ is a second free parameter modeling the sensitivity of the decision maker to utility differences ($\beta \geq 0$).

Our goal here is to show that by applying LOTO, we can extract information about parameter $\kappa$ (which we assume to vary from trial to trial) that goes beyond the information already provided by the observations. We do this by looking at the first derivatives of the log-likelihood of the HD model with respect to $\kappa$. Let us assume that option $i$ is an 'immediate' option, offering amount $x_i$ at delay $d_i = 0$, so that $u_i = x_i$, and that option $j$ is a 'delayed' option, offering amount $x_j$ ($x_j > x_i$) at delay $d_j$ ($d_j > 0$). It turns out that the first derivative of the log-likelihood for choosing the immediate option is:

$$\frac{\partial LL_i}{\partial \kappa} = \frac{\beta * x_j * d_j}{\left[1 + \kappa * d_j\right]^2} * (1 - p_i).$$  (8a)

Detailed derivations are provided in the 'Materials and methods'. Similarly, the first derivative of the log-likelihood for choosing the delayed option is:

$$\frac{\partial LL_j}{\partial \kappa} = -\frac{\beta * x_j * d_j}{\left[1 + \kappa * d_j\right]^2} * p_i.$$  (8b)

Note that the MLE can be obtained by setting these derivatives to 0. Importantly, we see that in contrast to the derivatives of the binomial distribution, the derivatives of the HD model depend on features of the task, namely on the amount and delay of the delayed option (i.e., $x_j$ and $d_j$) and on the amount of the immediate option (i.e., $x_i$), which is hidden in $p_i$. At first glance, this already looks as if the estimation of single-trial values of $\kappa_t$ will vary with the task's features, which is what we desired. Note, however, that both of the terms on the right-hand side of the derivatives are $\geq 0$.

Thus, by setting the derivatives in *Equation 8a* and *Equation 8b* to 0 to obtain the MLEs, parameter $\kappa_t$ will take the highest possible value (i.e., $\kappa_t = 1$) whenever the immediate option is chosen and the lowest possible value (i.e., $\kappa_t = 0$) whenever the delayed option is chosen. (Intuitively speaking: the best account for choosing the immediate option is to assume maximal impatience, the best account for choosing the delayed option is to assume is maximal patience.) This is illustrated in *Figure 2A and B*, which depict the log-likelihoods of choosing $i$ (left panels) or $j$ (middle panels) and their derivatives as a function of $\kappa$ for two different trials with $x_{i,1} = 20$, $x_{j,1} = 50$, and $d_{j,1} = 20$ in trial 1 (bright colors) and $x_{i,2} = 18$, $x_{j,2} = 100$, and $d_{j,2} = 25$ in trial 2 (dark colors). Independent of the features of the task (i.e., the trial-specific amount and delays), the log-likelihoods of choosing $i$ are monotonically increasing, and their derivatives are always positive, whereas the log-likelihoods of choosing $j$ are monotonically decreasing, and their derivatives are always negative.

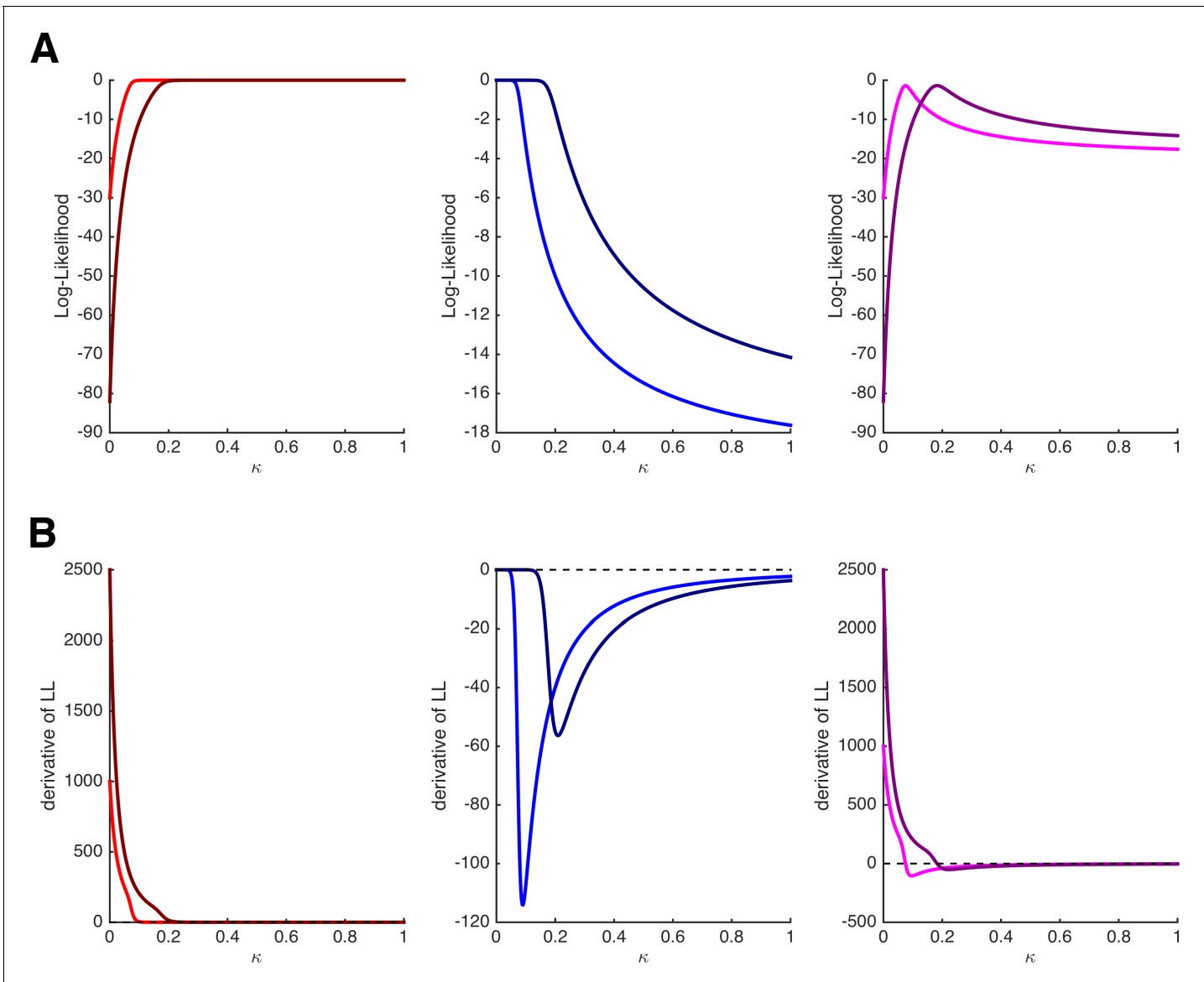

**Figure 2.** Log-likelihoods and derivatives for the hyperbolic discounting model. (A) Log-likelihoods for choosing the immediate option (left panel), choosing the delayed option (middle panel), and choosing the immediate option once and the delayed option once (right panel) as a function of parameter $\kappa$. The bright and dark colors refer to two different trials with different amounts and delays. (B) First derivatives of the log-likelihoods shown in panel A. Note that the estimation of $\kappa$ will depend on the features of the task only when both options are chosen at least once.
DOI: https://doi.org/10.7554/eLife.42607.003

Critically, however, this caveat disappears as soon as each option ($i$ and $j$) is chosen at least once. For instance, when keeping $x_i$, $x_j$, and $d_j$ constant over two trials in which both $i$ and $j$ are chosen once, the derivative of the log-likelihood becomes:

$$\frac{\partial LL_{\{i,j\}}}{\partial \kappa} = \frac{\partial LL_i}{\partial \kappa} + \frac{\partial LL_j}{\partial \kappa} = \frac{\beta * x_j * d_j}{\left[1 + \kappa * d_j\right]^2} * (1 - 2 * p_i),$$ (8c)

which is positive if $p_i < .5$ but negative if $p_i > .5$. The right panels of *Figure 2A and B* show the log-likelihoods and their derivatives when assuming that trial 1 or trial 2 were presented twice, and $i$ was chosen in one occasion and $j$ was chosen in the other occasion. Here, the derivatives cross the 0-line, and even more importantly, they do so at different values of $\kappa$ for the different trials (the MLE of $\kappa$ for trial 2 is higher because the immediate option is comparatively less attractive than in trial 1).

Why does this allow LOTO to extract more information about $\kappa_t$ than that given by the observations? Recall that the estimate of LOTO is the difference between the MLE of $\kappa$ for all trials and the MLE for all trials except $t$. As explicated above, these two MLEs will depend on the task features in the $n/n–1$ trials as long as both choice options (e.g., immediate, delayed) have been chosen at least once in the $n/n–1$ trials (so that we get non-extreme estimates of $\kappa$ from the $n/n–1$ trials). In principle, LOTO can therefore be applied as soon as both options have been chosen at least twice over the course of an experiment (if one option has been chosen in only one trial, then the estimate of $\kappa$ when excluding this particular trial will be extreme as in the single-trial case). We illustrate this using the two example trials from *Figure 2* at the end of the 'Materials and methods'.

It follows that LOTO should outperform the arguably most elementary method of capturing trial-by-trial parameter variability, which we denote 'single-trial fitting'. With single-trial fitting, we mean that after estimating all parameters of a model once for all trials, a specific parameter of interest (e.g., $\kappa$ of the HD model) is fitted again to each trial individually (while all other parameters are fixed to their 'all-trial' estimates). Here, we show that LOTO but not single-trial fitting of parameter $\kappa$ of the HD model is able to extract additional information about the true $\kappa_t$, over and above what is already provided by the observations. Thereto, we generated 160 trials with a constant immediate choice option and variable delayed choice options, and simulated decisions of a participant with $\kappa_t$ varying from trial to trial. Then, we applied LOTO and the single-trial fitting method to recover $\kappa_t$. As can be seen in *Figure 3*, the estimates of LOTO (left panel) but not of single-trial fitting (right panel) are correlated with the true $\kappa_t$ after accounting for the observations (i.e., whether the immediate or the delayed option was chosen). In other words, the positive correlation between $\kappa_t$ and the single-trial estimates (dashed magenta line) can be attributed entirely to the fact that higher values of $\kappa_t$ lead to more choices of the immediate option, but the positive correlation between $\kappa_t$ and the LOTO estimates is further driven by how expectable or surprising a decision is (given the trial features $x_{i,t}$, $x_{j,t}$, and $d_{j,t}$).

To summarize, one approach to extract information about trial-by-trial changes in a parameter (over and above what is already provided by the observations) is to introduce variations in the experiment that are linked in a systematic manner to the cognitive model. Because its estimates are not solely dependent on the behavior in a single trial but also take all remaining trials into account, LOTO allows to capture this additional information. Note, however, that for specific models, such as sequential sampling models of decision making, LOTO can be useful even without introducing task variations. In the next section, we will look at further improvements in LOTO's ability to recover trial-specific parameter values by extending the HD model to allow the prediction of choice and RT conjointly.

## IV. LOTO benefits from modeling an enriched set of data

Using LOTO to recover trial-specific parameter values of models that predict only binary outcomes, such as choices or accuracy rates, is possible but somewhat limited. This can be seen when looking at the left panel of *Figure 3*, which suggests that LOTO's estimates are always higher when the immediate option was chosen. The reason for this is that taking one trial out in which the immediate/delayed option was chosen will always de-/increase the estimation of the discount factor $\kappa$, because (as stated above) the single-trial log-likelihood is monotonically in-/decreasing given the choice. (The magnitude of this change varies as a function of the trial-specific choice options, but the sign of this change is determined by the choice.) However, this strict choice-dependent separation

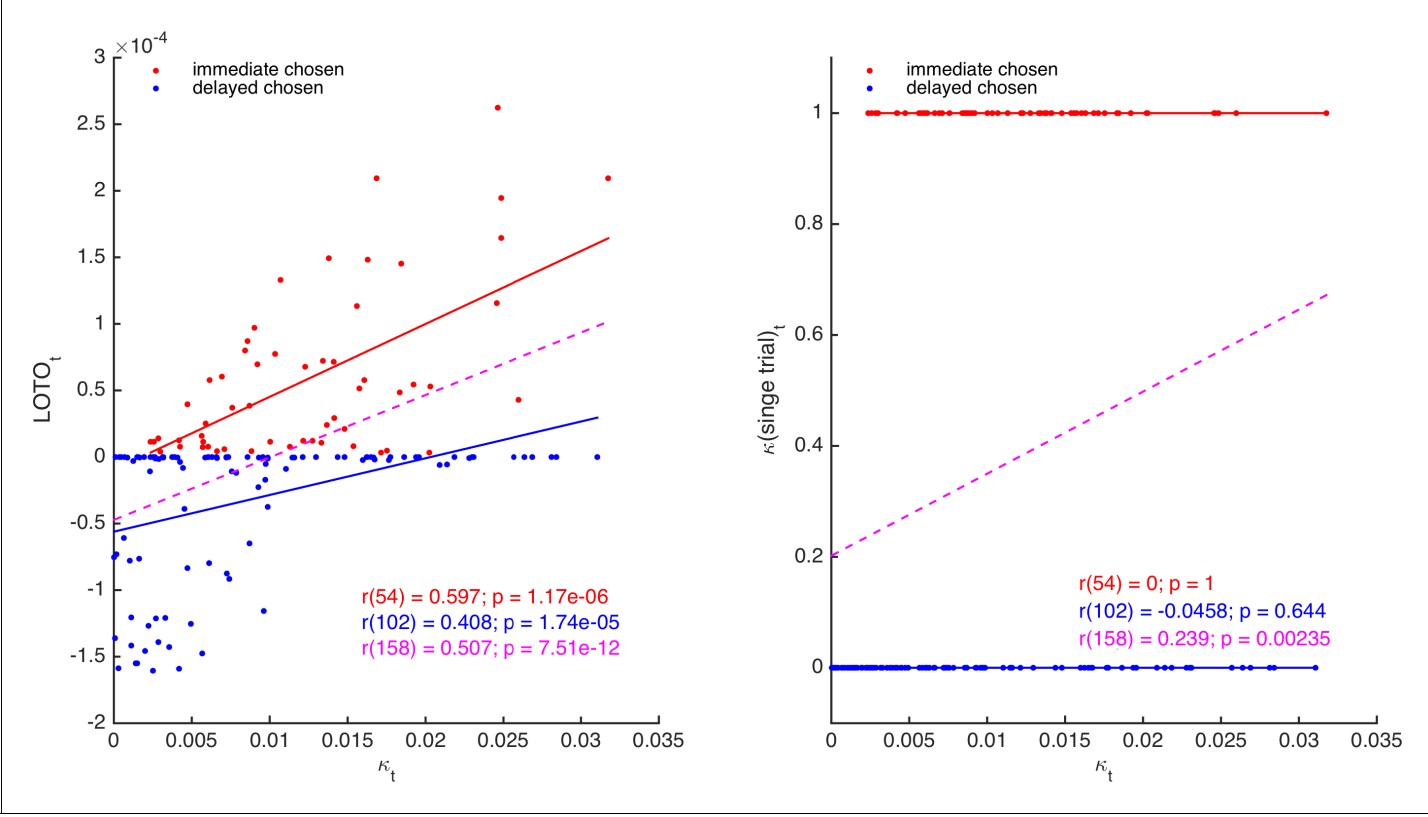

**Figure 3.** Comparison of LOTO and single-trial fitting. The left panel shows the correlation of the trial-specific parameter $\kappa_t$ of the HD model with LOTO's estimates, separately for choosing the immediate (red) and for choosing the delayed (blue) option. The dashed line (magenta) shows the regression slope when collapsing over all trials. The right panel shows the same for the single-trial-fitting approach.

DOI: https://doi.org/10.7554/eLife.42607.004

might not be plausible; for example, it could very well be that the delayed option is chosen in a trial in which this option offered a very high reward $x_j$ with a very low delay $d_j$, even though the actual discount factor in that trial (i.e., $\kappa_t$) was higher than on average. To overcome this limitation, we suggest that the model is extended so that it can take RT data into account. Going back to the example, the intuition is that even if the highly attractive delayed option is chosen, a high discount factor $\kappa_t$ will make this decision more difficult, which will result in a comparatively long RT. Applying LOTO to the extended model then allows to capture this RT-specific information.

We extended the HD model by adding the LBA model to it (using an approach that is similar but not equivalent to that used by *Rodriguez et al., 2014* and *Rodriguez et al., 2015*). Specifically, we assume that the difference in utility between the immediate and delayed options is mapped onto their LBA's expected drift rates as follows:

$$\nu_i = \frac{1}{2} + \lambda * (u_i - u_j), \tag{9a}$$

$$\nu_j = 1 - \nu_i = \frac{1}{2} + \lambda * (u_j - u_i), \tag{9b}$$

where $\lambda$ is a free parameter that scales the relationship between utility and expected drift rate. Note that these definitions ensure that the two expected drift rates sum up to 1, following the original specification of the LBA (*Brown and Heathcote, 2008*). More specifically, $\nu_i$, and $\nu_j$ represent the means of two independent normal distributions (with standard deviation s), from which the actual drift rates in a trial are assumed to be drawn (see 'Materials and methods'). Choices and RT were simulated for 160 trials of 30 participants and LOTO was applied to the HD and the HD-LBA models.

Single-trial fitting was also applied to the HD-LBA model. *Figure 4A* depicts correlations of an example participant between the true $\kappa_t$ and LOTO's estimate for HD (left panel) and for HD-LBA (right panel). For HD-LBA, the correlation is much improved, and LOTO's estimates do not strictly separate choices of the immediate and delayed options anymore. The improved performance of LOTO is confirmed when comparing the correlations of all 30 simulated participants (*Figure 4C*; immediate chosen: $t(29) = 10.79$, $p < .001$, Cohen's $d = 1.97$, Bayes Factor in favor of the alternative hypothesis $BF_{10} > 1,000$; delayed chosen: $t(29) = 8.79$, $p < .001$, $d = 1.61$, $BF_{10} > 1,000$).

Notably, the single-trial fitting method applied to the HD-LBA model is also able to capture some variability over and above the information that is already contained in the decisions (*Figure 4*). This is because RT (and the interplay between RT and decisions) add further information about $\kappa_t$ that can be picked up by single-trial fitting. Nonetheless, applying LOTO to HD-LBA yields significantly

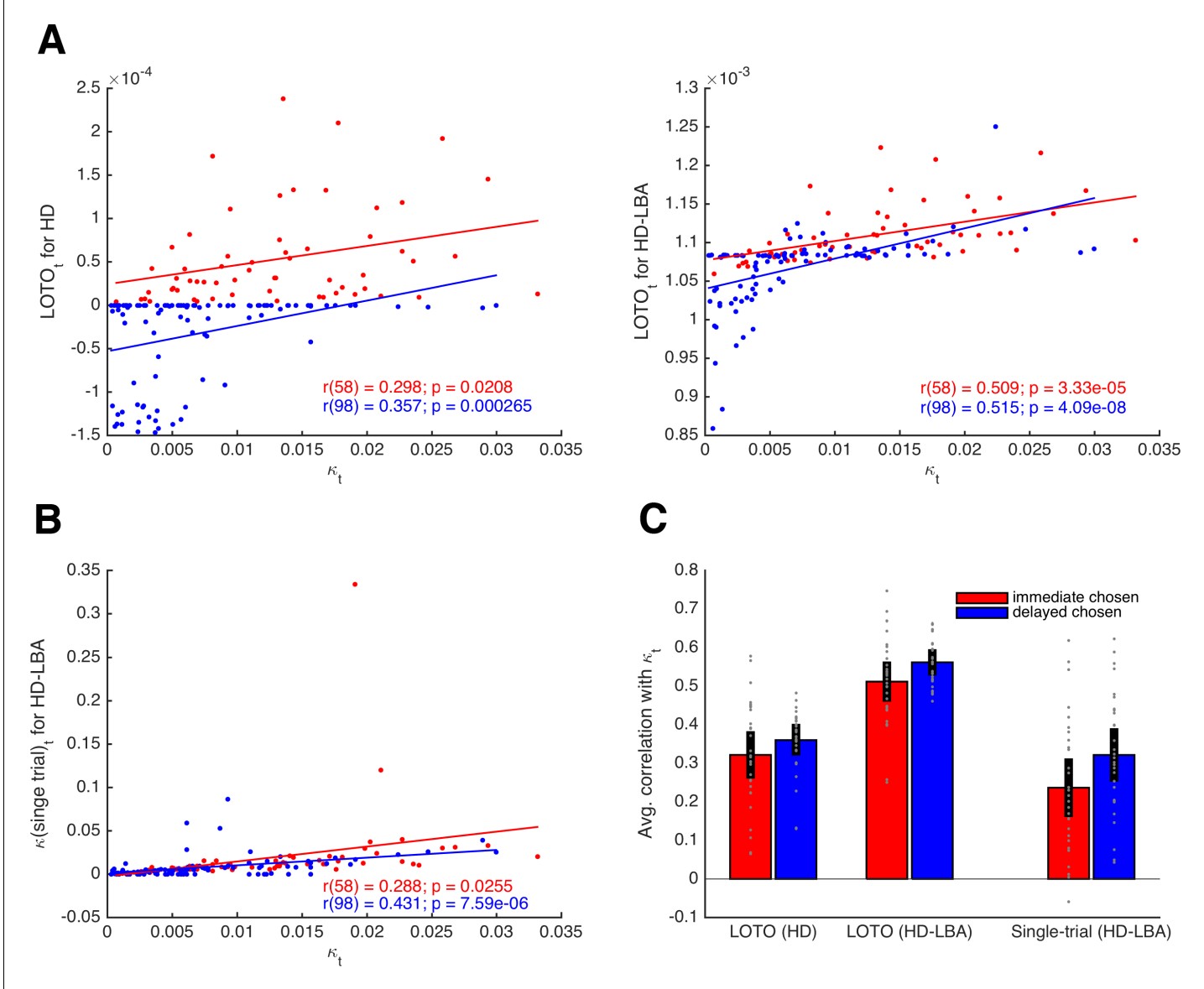

**Figure 4.** Comparison of LOTO for the HD and the HD-LBA models. (**A**) Correlation of the trial-specific parameter $\kappa_t$ with LOTO's estimates for an example participant. The left panel shows the results when using the HD model, the right panel when using the HD-LBA model. (**B**) Same as panel (**A**) but for the single-trial fitting method when using the HD-LBA model. (**C**) Average correlation for all 30 simulated participants. Individual values (gray dots) are shown together with the 95% confidence intervals of the mean (black error bars).
DOI: https://doi.org/10.7554/eLife.42607.005

better recovery of the trial-wise parameter values (*Figure 4C*; immediate chosen: $t(29) = 7.24$, $p < .001$, $d = 1.32$, $BF_{10} > 1,000$; delayed chosen: $t(29) = 7.71$, $p < .001$, $d = 1.41$, $BF_{10} > 1,000$). Taken together, recovering trial-by-trial changes in a parameter of a cognitive model with LOTO (but also with other techniques) is much more efficient when the model is able to predict a rich set of single-trial observations, such as RT (and possibly even continuous ratings of choice confidence as in *Pleskac and Busemeyer, 2010*) in addition to choices or accuracy rates.

## V. A non-parametric test for the presence of systematic trial-by-trial variability

So far, our derivations and simulations have rested on the assumption that systematic variability in the parameter of interest exists. As we have argued before, however, this assumption does not have to be correct — not even when the overt behavior appears to be stochastic (*Gluth and Rieskamp, 2017*). It could be that stochasticity is induced by random fluctuations that are not accounted for by the cognitive model (e.g., randomness in motor execution). When assuming variability in a specific parameter, the risk is that such unsystematic variability is taken up by LOTO even in the absence of any systematic variability in the parameter. Hence, the question is whether one can separate situations in which LOTO captures systematic vs. unsystematic trial-by-trial variability.

To test for the presence of systematic trial-by-trial variability, we propose a non-parametric and simulation-based approach that exploits the improvement in model fit by LOTO: if a model is fitted to $n-1$ trials, the fit statistic (e.g., the deviance) will improve for two reasons: i) the statistic for the left-out trial is excluded; and ii) the parameter estimate is adjusted so that it optimally fits the $n-1$ dataset. The rationale of our proposed approach is that the improvement in fit that is attributable to the second reason (i.e., the adjustment) will be lower when there is no systematic trial-by-trial variability in the parameter of interest compared to when there is systematic variability. Practically, we suggest conducting this test in five steps:

1. For each participant, estimate parameters of the cognitive model for all trials and compute the goodness-of-fit per trial.
2. For each participant, generate many (e.g., 1,000) simulations of the model for all trials and all participants using the participants' estimated parameters and assuming no variability in the parameter of interest (i.e., the null hypothesis).
3. Estimate parameters for these simulated data again, apply LOTO, and compute the LOTO-based improvement in goodness-of-fit for the $n-1$ trials.
4. Do the same as in step 3, but for the empirical data.
5. Test whether the LOTO-based improvement in fit for the empirical data is greater than 95% of the improvements in the simulations that were generated under the null hypothesis.

Note that one advantage of this non-parametric approach is that it does not rely on any assumptions about how the improvement of model fit is distributed under the null hypothesis.

*Figure 5A* shows the results of this test for the HD model introduced in 'Results: Section III'.

We generated 1,000 simulations per participant under the null hypothesis and 10 additional simulations with increasing levels of variability in parameter $\kappa$ (for details, see 'Materials and methods'). Importantly, the figure shows the average difference in deviance between the 'all-trial' fit and the LOTO fits only for the $n-1$ trials that were included in each LOTO step. This means that the trivial improvement in fit resulting from the fact that LOTO has to explain one trial less is taken out. Thus, the histogram in *Figure 5A*, which shows the results for the 1,000 simulations under the null hypothesis, confirms our suspicion that LOTO will improve the fit even in the absence of true trial-by-trial variability in parameter $\kappa$. However, the continuous vertical lines, which show the results for the simulations with true variability of increasing amount (from blue to red), indicate significantly higher improvements in fit when variability is actually present — except for the lowest level of variability (the dashed black line indicates the 95% threshold).

What happens if there is no systematic variability in the parameter of interest (here: $\kappa$), but there is systematic variability in another parameter (here: $\beta$)? We can repeat the previous analysis but now simulate variability (of increasing amount) in $\beta$ instead of $\kappa$. (Note that variability in $\beta$ would mean that the utility difference between choice options is mapped onto the decision in a more/less deterministic way, depending on whether $\beta_t$ is high or low in a trial; for instance, choosing the immediate option in a trial in which the delayed option is very attractive could be attributed to a low $\beta_t$

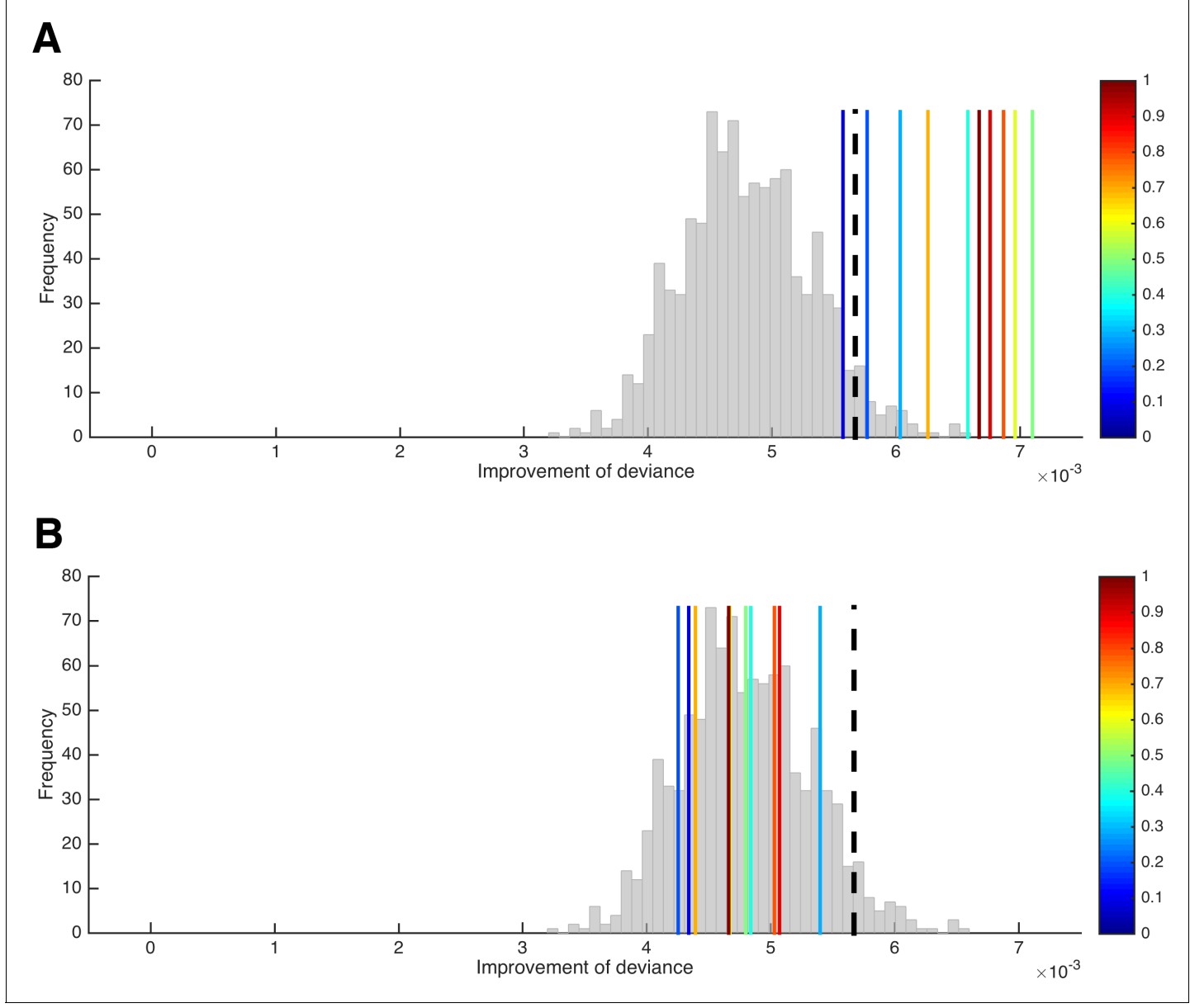

**Figure 5.** A test for the presence of parameter variability. (**A**) Improvement of the HD model's deviance (in the *n*–1 trials included per LOTO step) from the 'all-trials' fit to the LOTO fit when variability in κ is absent (histogram) and when variability in κ of increasing amount is present (blue to red vertical lines). The black dashed line indicates the 5% of simulations under the null hypothesis with the largest improvement and serves as the statistical threshold. (**B**) Same as in panel **A** but with variability in *β* instead of κ.

DOI: https://doi.org/10.7554/eLife.42607.006

The following figure supplements are available for figure 5:

**Figure supplement 1.** Testing for misattribution of variability in input and output variables to variability in a parameter.
DOI: https://doi.org/10.7554/eLife.42607.007
**Figure supplement 2.** Testing for the presence of parameter variability and for misattributions in an adapted task.
DOI: https://doi.org/10.7554/eLife.42607.008

value instead of a high $κ_t$ value.) As can be seen in *Figure 5B*, the improvement in goodness-of-fit when trying to explain variability in *β* by modeling variability in κ using LOTO does not exceed the improvement under the null hypothesis, and there is no systematic relationship with the amount of variability in *β*. Thus, we can conclude that finding a significantly higher improvement in fit in our data compared to simulated data under the null hypothesis cannot be due to a misattribution of

variability in $\beta$ to variability in $\kappa$. (Of course, this conclusion does not have to hold for other models, parameter sets, and tasks; the test has to be conducted anew for every different situation.)

Variability in behavior (and neural activity) may derive not only from variability in model parameters but also from the processes used to infer the input and output signals of a computation (*Drugowitsch et al., 2016*; *Findling et al., 2018*; *Loomes et al., 2002*; *Polanía et al., 2019*). For example, choosing the immediate or delayed option in the intertemporal choice task might depend not only on the current impulsiveness (i.e., $\kappa$) but also on whether the utility of the delayed option (i. e., $u_j$) is computed correctly or whether the amount $x_j$ and the delay $d_j$ are perceived accurately. Thus, it might be desirable to examine whether LOTO can dissociate potential variability in these features from variability in the parameter of interest.

Accordingly, we simulated data of the HD model without trial-by-trial variability in $\kappa$ but with variability in either the amounts $x_j$, the delays $d_j$ or the computation of utilities $u_j$ of the delayed option. (Note that we ensured that the degree of variability in these variables was similar to that of $\kappa$ in the previous simulations; see 'Materials and methods'.) Then, we applied LOTO to these simulated data to test whether LOTO misattributes the variability to $\kappa$. As can be seen in *Figure 5—figure supplement 1*, LOTO does not misattribute variability in amounts, but for 3 out of 10 simulations of delay-based variability and for 4 of 10 simulations of utility-based variability, a misattribution could not be ruled out. This appears reasonable given the tight connection of the parameter to both delays and utilities (see *Equation 6*).

How can the separability of the discount factor $\kappa$ from the input and output variables $x$, $d$, and $u$ be improved? We reasoned that changing the intertemporal choice task by increasing the number of options should help: whereas the discount factor can be assumed to be stable across options (and to vary only from trial to trial), the variability in the input and output variables can be assumed to differ between options (i.e., they are drawn from independent distributions for each option). Accordingly, we extended the simulated task to three options, which can still be considered a reasonable experimental design (e.g., *Gluth et al., 2017*). We then performed the test for systematic variability using simulated data for this extended task that exhibited true variability in either $\kappa$, $x$, $d$, or $u$. As predicted, a significant improvement in model fit over and above the improvement under the null hypothesis of no variability was only found for $\kappa$ but not for any of the input or output variables (*Figure 5—figure supplement 2*). In conclusion, LOTO might be an efficient tool to dissociate parameter variability from variability in the input and output signals of computational processes, assuming an appropriate experimental design and careful simulations.

## VI. Inferring trial-by-trial variability in more than one parameter

An advantageous feature of LOTO is that the duration of applying the method is almost independent of the number of parameters for which trial-by-trial variability is inferred. The full model is still estimated once for all $n$ trials, and LOTO is still applied $n$ times (but the estimation might take a bit longer for multiple parameters). In this section, we illustrate the application of LOTO to inferring trial-by-trial variability in two parameters of the HD-LBA model: the discount factor $\kappa$ and the decision threshold $b$ (for details, see 'Materials and methods').

First, we used the non-parametric test from the previous section to check for the presence of systematic variability. In doing this, we simulated 1,000 sets of 30 participants under the null hypothesis and tested whether the simulation with true variability in $\kappa$ and $b$ provides a significantly improved deviance than those simulations. This was confirmed as only 19 of the 1,000 simulations under the null hypothesis provided a better fit. It should be noted that the explanatory power of this test is somewhat limited, because we can only conclude that there is systematic variability in at least one of the two parameters but not necessarily in both of them.

Next, we looked at the correlations between the true $\kappa_t$ and $b_t$ parameter values and their LOTO estimates for an example participant. As shown in the upper left panel of *Figure 6A*, LOTO is still capable of recovering the variability in parameter $\kappa$ from trial to trial for both types of decisions (i.e., immediate and delayed). Similarly, LOTO captures the variability in parameter $b$ (lower right panel). The lower left and upper right panels in *Figure 6A* show the correlations between the true $\kappa_t$ and $b_t$ with the 'wrong' LOTO estimates of $b_t$ and $\kappa_t$, respectively. Ideally, one would desire correlations close to zero here, that is, LOTO should not attribute variability in one parameter to variability in another parameter. However, we see that there are modest correlations whose signs depend on the choice. When the immediate option was chosen, the correlations are negative; when the delayed

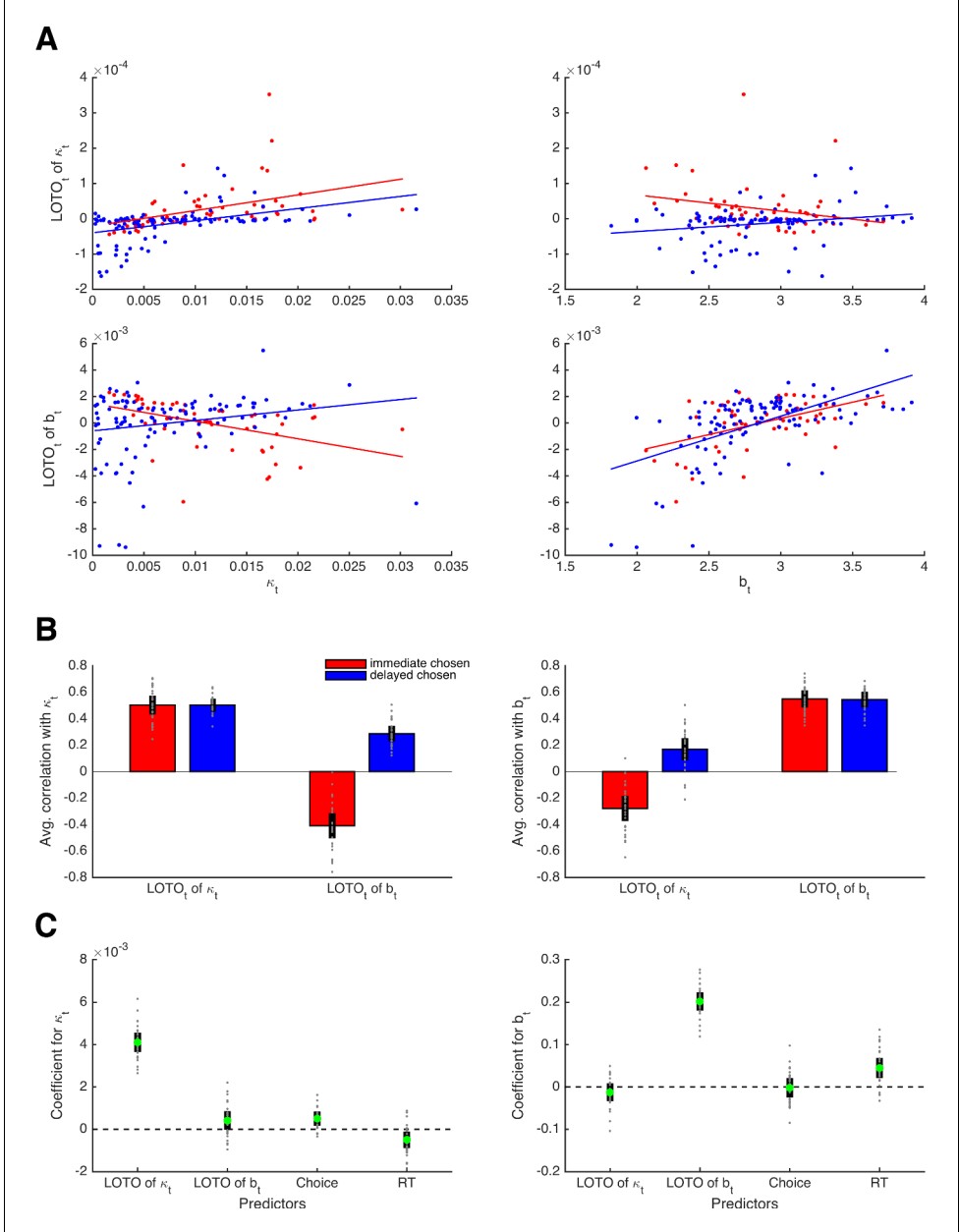

**Figure 6.** Capturing trial-by-trial variability in two parameters. (A) Correlations of true parameter values $\kappa_t$ and $b_t$ with their LOTO estimates for an example participant; the on-diagonal elements show the 'correct' inferences, the off-diagonal elements show the 'wrong' inferences. (B) Average correlations for all simulated participants (left panel: correlation with $\kappa_t$; right panel: correlation with $b_t$). (C) Average standardized regression coefficients of LOTO estimates, choices, and RT when explaining variance in $\kappa_t$ (left panel) and $b_t$ (right panel). Individual values (gray dots) are shown together with 95% confidence intervals of the mean (black error bars).

DOI: https://doi.org/10.7554/eLife.42607.009

option was chosen, the correlations are positive. These results are confirmed when looking at the average correlations across all 30 simulated participants (*Figure 6B*): LOTO attributes variability in the 'correct' parameters independent of choice and variability in the 'wrong' parameters depending on which option was chosen. (Intuitively, a high discount factor $\kappa_t$ competes with the decision threshold $b_t$ as an alternative explanation for fast choices of the immediate option or slow choices of the delayed option. Similarly, a high decision threshold $b_t$ competes with the discount factor $\kappa_t$ as an alternative explanation for choices of the option with highest utility.)

Importantly, these results indicate that the observations (i.e., choices and RT) could explain away the undesired correlations while not affecting the desired correlations. To test this, we conducted random-effects linear regression analyses with the true parameter ($\kappa_t$, $b_t$) as the dependent variable and with the LOTO estimates of these parameters together with choices and RT as independent variables (*Figure 6C*). The standardized regression coefficient (averaged across simulated participants) for LOTO's estimate of $\kappa_t$ when explaining variance in true $\kappa_t$ was indeed significantly positive with a very high effect size ($t(29) = 28.48$, $p < .001$, $d = 5.20$, $BF_{10} > 1,000$). The coefficient for the 'wrong' estimate of $b_t$ when explaining variance in $\kappa_t$ also reached significance, but the effect was much weaker ($t(29) = 3.01$, $p = .005$, $d = 0.55$, $BF_{10} = 7.74$). With respect to explaining variance in true $b_t$, only LOTO's estimate of the 'correct' estimate of $b_t$ was significant and strong ($t(29) = 30.40$, $p < .001$, $d = 5.55$, $BF_{10} > 1,000$), whereas the effect for the 'wrong' estimate of $\kappa_t$ did not reach significance ($t(29) = -1.93$, $p = .063$, $d = -0.45$, $BF_{10} = 0.99$).

Taken together, LOTO makes it possible to infer trial-by-trial variability in more than one parameter within the same model, revealing information over and above that provided by the observations. At the same time, LOTO's misattributions of variability to other parameters are largely explained away by the observations. (As before, these conclusions are restricted to the current setting of task, model, and parameters, and have to be tested anew for different settings.) Notably, the random-effects linear regression analysis conducted above is very similar to the conventional approach of analyzing fMRI data on the basis of the General Linear Model (*Friston et al., 1994*). The ultimate question is not whether LOTO induces some significant misattribution or not, but whether this misattribution can in turn be expected to lead to wrong inferences about the relationship between brain activity and variability in cognitive parameters. We further elaborate on this issue in 'Results: Section X'.

## VII. The performance of LOTO as a function of the number of trials

Trial numbers in neuroimaging studies are often low. fMRI studies are particularly limited because scanning hours are expensive, and because the low temporal resolution of fMRI requires long inter-trial intervals in the case of event-related fMRI (which is most suitable for trial-by-trial analyses). Therefore, it is important to understand the extent to which LOTO's accuracy depends on the number of trials. To test this, we repeated the simulations of the HD and the HD-LBA models for 20, 40, 80, 160, 320, and 640 trials. As shown in *Figure 7A*, the average extent of the correlations between the true parameter values and the respective estimates of LOTO are largely unaffected by the number of trials. The recovery performance is slightly reduced for the threshold parameter $b$ of the HD-LBA model in the case of 20 trials. Note, however, that the variance of the correlations decreases as the number of trials increases. Therefore, LOTO still benefits from increasing the number of trials in terms of achieving higher statistical power, as we show in 'Results: Section X'.

In 'Results: Section II', we mentioned that the impact of a single trial on the overall model fit, as well as on the LOTO estimate, decreases as the number of trials increases. If the number of trials is very high, then the improvement in model fit that results from a change in the parameter value might become smaller than the default tolerance criterion set by the minimization algorithm. In other words, the algorithm assumes convergence because the change in model fit caused by taking one trial out is insignificantly small. *Figure 7B* illustrates this phenomenon. The upper panels show the results for parameter $\kappa$ of the HD model for 160 trials, the lower panels for 1,600 trials. In the left panels, a strict tolerance criterion was used; in the right panels, the default tolerance criterion in MATLAB was used (for details, see 'Materials and methods'). In the case of 1,600 trials, the tolerance criterion matters: if the default criterion is used, the LOTO estimates exhibit discrete jumps, that is, the same parameter value is estimated for trials with relatively similar observations, because a more fine-graded estimation does not improve the fit beyond the default tolerance criterion. This is not the case for the strict tolerance criterion.

According to the example in *Figure 7B*, the default tolerance criterion appears to lower the correlation between the true values and the LOTO estimates in the case of 1,600 trials, but only to a minor extent. This observation is confirmed when comparing the correlations across all 30 simulated participants. In the case of 160 trials, the correlations obtained using the strict and the default tolerance criterion are not significantly different ($t(29) = 1.27$, $p = .213$, $d = 0.23$, $BF_{10} = 0.40$). In the case of 1,600 trials, there is a significant difference ($t(29) = 4.59$, $p < .001$, $d = 0.84$, $BF_{10} = 323.03$), but the difference is small in absolute terms (i.e., on average the correlation is reduced by only .003). In

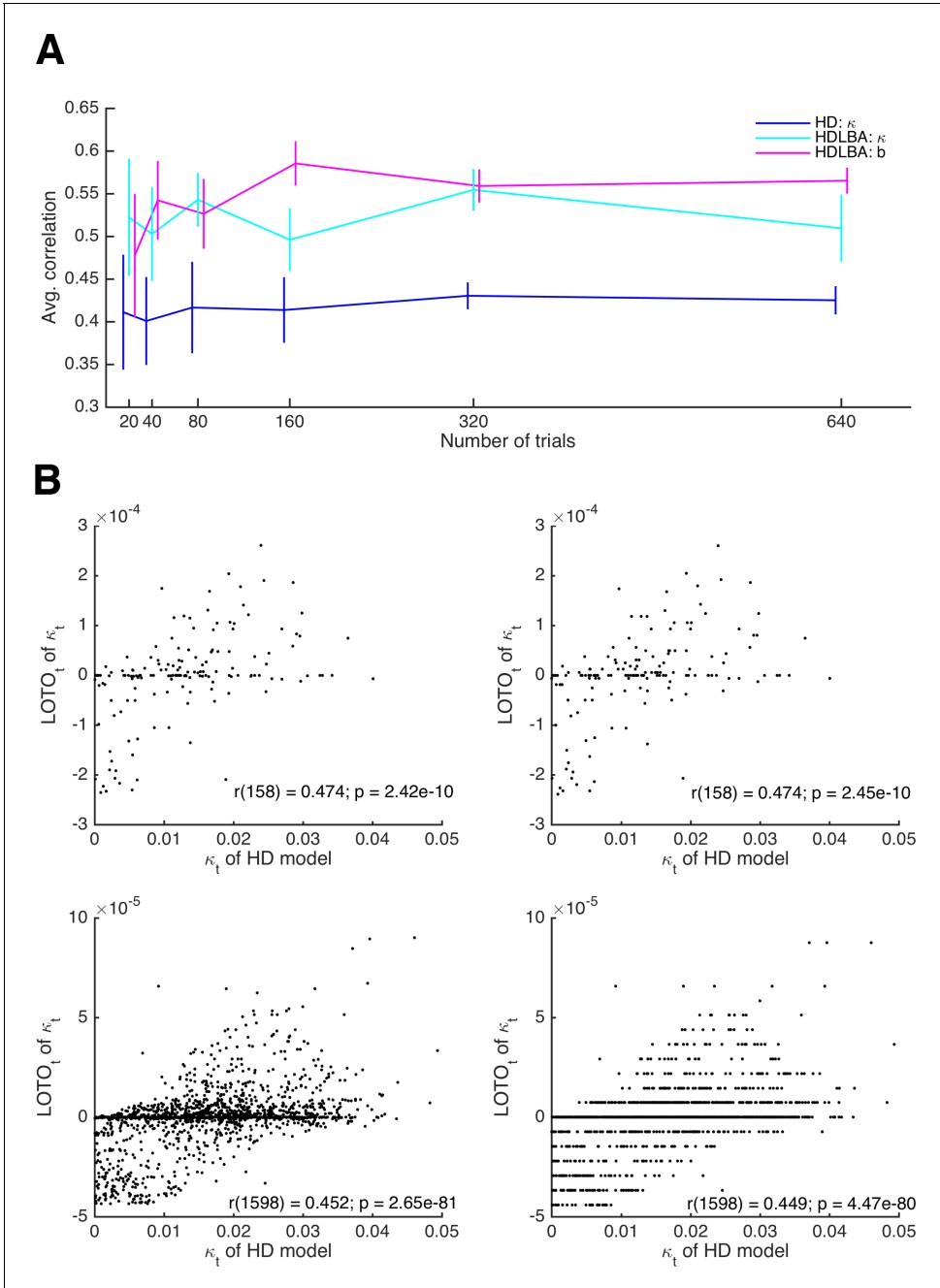

**Figure 7.** Relationship between the performance of LOTO and the number of trials. (**A**) Average correlations between true parameter values and LOTO estimates for parameters of the HD and the HD-LBA models as a function of the number of trials per simulated participant. (**B**) Example correlations between true values and LOTO estimates for 160 trials (upper panels) and for 1,600 trials (lower panels), separately for a strict tolerance criterion for convergence of the minimization algorithm (left panels) and for a more lenient, default tolerance criterion (right panels). Error bars indicate 95% confidence intervals of the mean.

DOI: https://doi.org/10.7554/eLife.42607.010

general, we recommend checking the potential impact of the tolerance criterion on LOTO's performance by running simulations, in particular when LOTO is applied to a task with many trials (i. e., $\geq 1{,}000$).

## VIII. Invariance of LOTO with respect to the underlying parameter distribution

In contrast to some other methods for inferring trial-by-trial variability in model parameters (*van Maanen et al., 2011*; *Turner et al., 2015*), LOTO does not require the assumption that parameters are drawn from specific distributions. Even though it is not the purpose of LOTO, one might wonder whether the distributions of parameters and of LOTO's estimates can be expected to be similar. To answer this question, we reverted to the initial 'toy' model example of the binomial distribution ('Results: Section II'). For this purpose, we sampled 10,000 Bernoulli events, which we split into 500 'blocks' of 20 trials each (see 'Materials and methods'). For each block, parameter $\theta_m$ was drawn from either a normal or a uniform distribution and observations $k_m$ were then drawn from binomial distributions given $\theta_m$. Finally, LOTO was applied to infer variability in $\theta$.

The left and right panels of *Figure 8A* depict the normally and uniformly distributed parameter values of $\theta_m$, respectively. Similarly, the left and right panels of *Figure 8B* show the distributions of the corresponding LOTO estimates. The distributions of the true parameter values are markedly different, whereas the distributions of LOTO estimates are very similar. The reason for this can be seen in *Figure 8C*, which depicts the distribution of observations $k_m$, which perfectly match the LOTO distributions (because the observations and LOTO estimates are perfectly correlated for this 'toy' model; see 'Results: Section II'). By definition, the observations are distributed binomially and so are the LOTO estimates. Generally speaking, the distribution of LOTO estimates is mostly invariant to the distribution of the underlying parameters but depends much more on how the parameter values map onto the observations via the model and its likelihood function.

To generalize these conclusions to the more interesting case of a cognitive model, we sampled single-trial parameter values of $\kappa$ (i.e., the discount factor) and $b$ (i.e., the decision threshold) of the HD-LBA model either from a normal distribution or from a uniform distribution (details are provided in 'Materials and methods'). The results for parameter $\kappa$ and for parameter $b$ are presented in *Figure 8—figure supplement 1* and *Figure 8—figure supplement 2*, respectively. Again, the distributions of LOTO estimates appear to be invariant to the shape of the underlying parameter distributions. In this case of the HD-LBA model, the parameter values and LOTO's estimates are linked via the LBA likelihood function. Most of all, this function depends on the drift rates that are drawn from normal distributions. Expectedly, the LOTO estimates appear to be roughly normally distributed. Importantly, our simulations of the HD-LBA model also suggest that the performance of LOTO (in terms of the correlation between true parameter values and LOTO estimates) does not seem to depend much on the distribution of the underlying parameter values. Hence, we conclude that LOTO cannot be used to infer the shape of the underlying parameter distribution. Despite this limitation, LOTO is highly efficient in detecting significant relationships between cognitive model parameters and neural data, regardless of the underlying parameter distribution.

## IX. Comparison of LOTO with related approaches

In this section, we compare LOTO's ability to recover the underlying trial-by-trial variability of cognitive model parameters with alternative techniques. More specifically, we generated synthetic data (i.e., choices and RT) from the LBA model (*Brown and Heathcote, 2008*), which assumes that the trial-wise drift rates $\nu_t$ and start points $A_t$ are drawn from normal and uniform distributions, respectively (details are provided in the 'Materials and methods'). We compare LOTO with three other methods: i) with our previously proposed technique (*Gluth and Rieskamp, 2017*; henceforth 'GR17'), ii) with 'single-trial fitting', and iii) with the 'Single-Trial LBA' approach (*van Maanen et al., 2011*; henceforth 'STLBA'), which was specifically developed for retrieving maximum-likelihood estimates of trial-wise drift rates and start points of the LBA.

*Figure 9* shows correlations between the true drift rates and start points and the values recovered by each method for an example simulation (upper panels), together with average correlations for multiple simulated experiments (lowest panels). From the example correlations, we see that LOTO and GR17 yield very similar results. This is to be expected because the two approaches employ similar principles. In both cases, parameters are estimated for all trials, and those parameters for which trial-by-trial variability is not assumed are fixed to their all-trial estimates; then, both approaches use the discrepancy between the observation in each trial and the observations over all trials to infer plausible trial-specific values. However, the comparison across multiple simulations

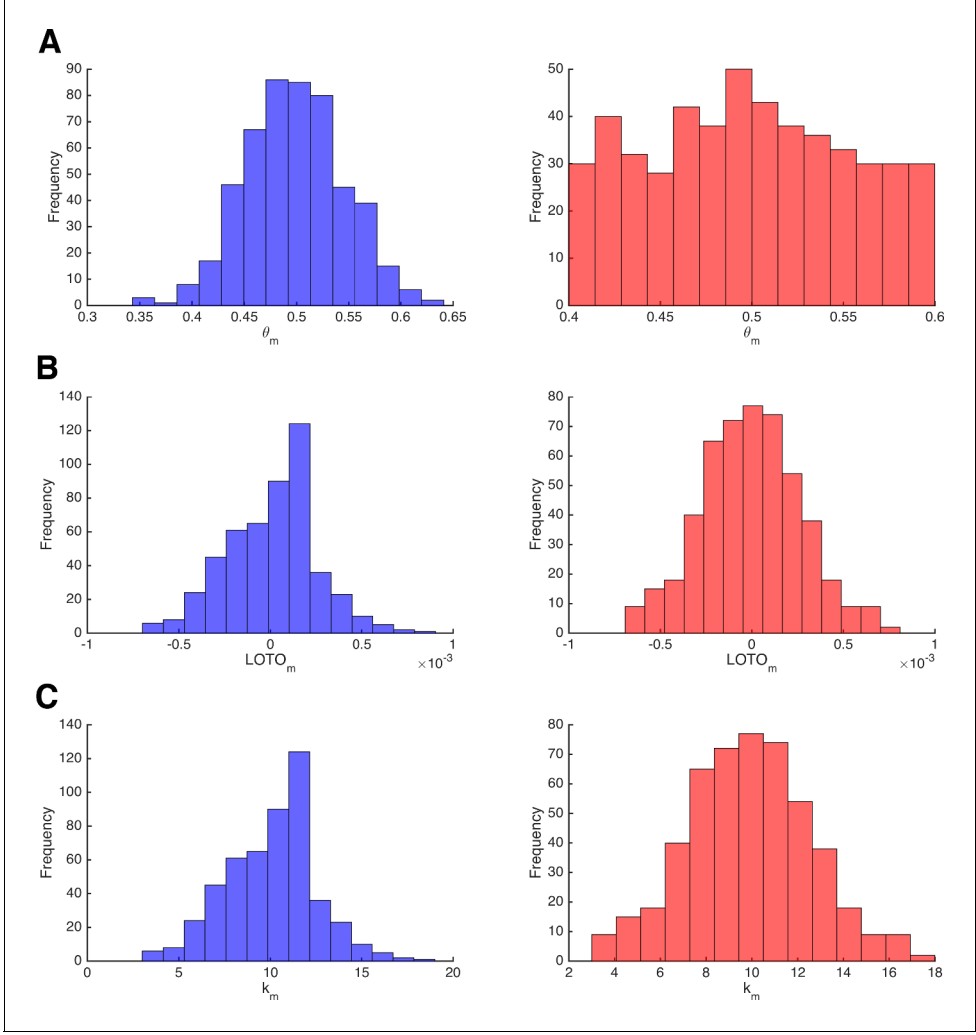

**Figure 8.** Invariance of LOTO with respect to the underlying parameter distribution. (**A**) Distribution of the data-generating parameter values $\theta_m$ for two simulations that sample from a normal distribution (left, blue) or from a uniform distribution (right, red). (**B**) The corresponding distributions of LOTO estimates. (**C**) The corresponding distributions of observations $k_m$.

DOI: https://doi.org/10.7554/eLife.42607.011

The following figure supplements are available for figure 8:

**Figure supplement 1.** Comparison of LOTO distributions for parameter $\kappa$ of the HD-LBA model.

DOI: https://doi.org/10.7554/eLife.42607.012

**Figure supplement 2.** Comparison of LOTO distributions for parameter $b$ of the HD-LBA model.

DOI: https://doi.org/10.7554/eLife.42607.013

reveals that LOTO performs significantly better than GR17 for both parameters (drift rate: $t(19)$ = 3.76, $p$ = .001, $d$ = 0.84, $BF_{10}$ = 28.75; start point: $t(19)$ = 4.86, $p < .001$, $d$ = 1.09, $BF_{10}$ = 259.17). The reason for this is that the performance of GR17 depends on the step size with which the range of possible parameter values is scanned, and we were forced to choose an immediate step size to avoid excessive computation times. As a consequence, the GR17 estimates exhibit discrete jumps (similar to those mentioned in the previous section when applying LOTO with a too lenient tolerance criterion), which lower the correlation with the true parameters. In other words, the advantage of LOTO over GR17 is its higher processing speed, which is particularly relevant when modeling variability in multiple parameters.

The 'single-trial fitting' approach yields acceptable results for the drift rate, but is still outperformed by LOTO ($t(19)$ = 19.72, $p < .001$, $d$ = 4.41, $BF_{10} > 1,000$). More dramatically, the method fails completely with respect to recovering the start point values. In our view, this is because

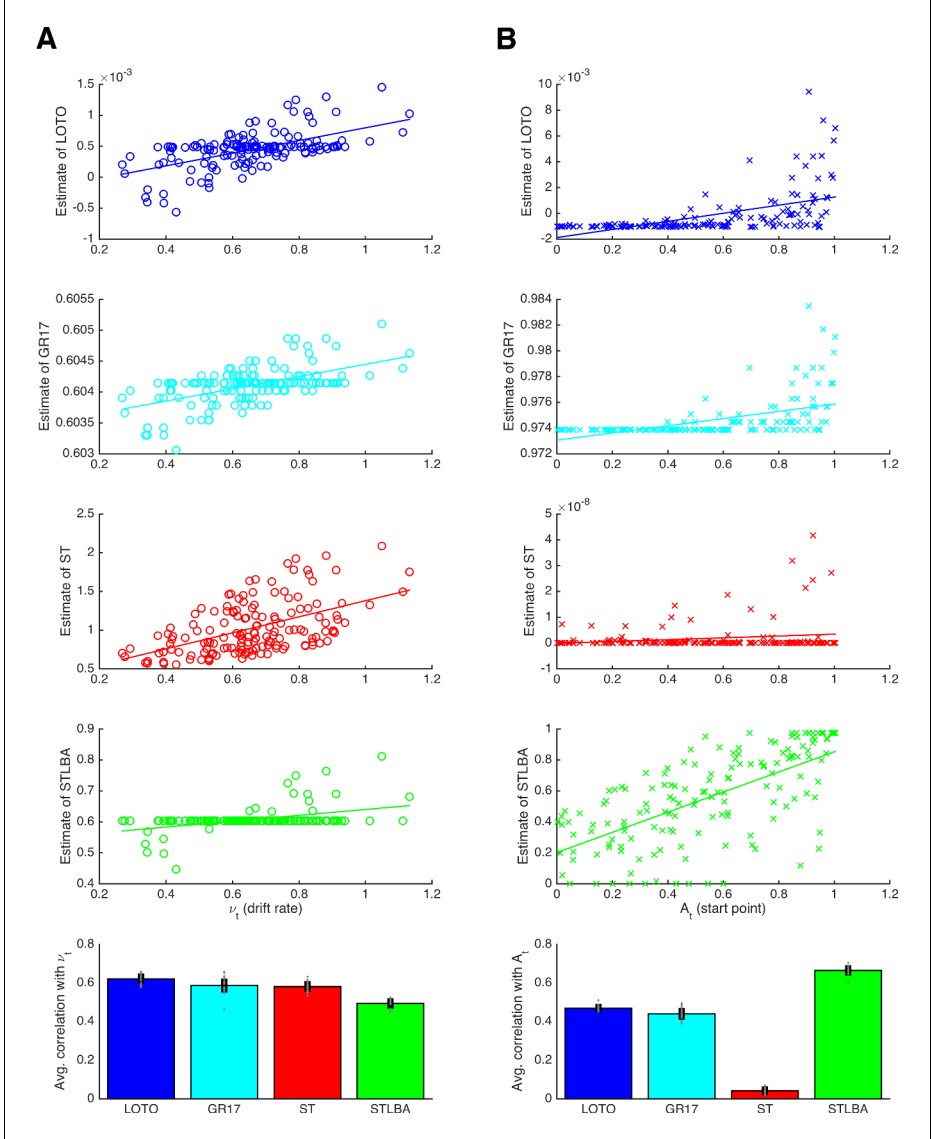

**Figure 9.** Comparison of methods for capturing trial-wise parameter values. (**A**) Results for the LBA drift rate parameter. The first four panels show correlations for an example participant for the methods LOTO (blue), GR17 (cyan), single-trial fitting (ST; red), and STLBA (green). The lowest panel shows correlations averaged over all simulations. (**B**) Same as panel **A** but for the LBA start point parameter. Individual values (gray dots) are shown together with 95% confidence intervals of the mean (black error bars).

DOI: https://doi.org/10.7554/eLife.42607.014

The following figure supplement is available for figure 9:

**Figure supplement 1.** Comparison of LOTO with Bayesian modeling and joint modeling.
DOI: https://doi.org/10.7554/eLife.42607.015

adjusting the drift rate is sufficient to account for the observations in a single trial (i.e., whether the choice was correct or not and how fast the choice was), so that there is no information left to be captured by the start point. By contrast, LOTO (and GR17) estimates depend on how the likelihood space of the LBA model changes when one trial is taken out, and this likelihood space depends on both parameters.

Finally, the STLBA method performs worse than LOTO with respect to the drift rate ($t(19) = 52.60$, $p < .001$, $d = 11.76$, $BF_{10} > 1,000$) but better with respect to the start point ($t(19) = -30.08$, $p < .001$, $d = -6.73$, $BF_{10} > 1,000$). Looking at the example simulations in **Figure 9**, we see that this is because many single-trial drift rates (but not start points) of the STLBA are equal to the 'all-trial

estimate'. This is a consequence of LBA's assumption that drift rates are drawn from a normal distribution whereas start points are drawn from a uniform distribution, which induces a 'ridge' in the joint density distribution on which one drift rate value (namely the mean of the normal distribution) together with many start points are the maximum-likelihood estimates (L. van Maanen; personal communication, February 2018). Overall, LOTO appears to provide comparatively good, and in fact mostly superior, results when compared to alternative approaches for capturing trial-by-trial variability in cognitive model parameters.

The previous comparison does not include arguably the most powerful approach for linking cognitive models with neural data, which is the joint modeling approach (*Turner et al., 2015*). This method employs hierarchical Bayesian modeling (*Farrell and Lewandowsky, 2018*; *Lee and Wagenmakers, 2013*) to draw both behavioral and neural parameters from a joint, higher-order probability distribution and to fit models to both behavioral and neural data. To compare LOTO to the joint modeling approach, we first implemented Bayesian modeling to obtain single-trial estimates of parameter $\kappa$ of the HD model (details are provided in the 'Materials and methods'). Then, we extended this to the joint modeling approach by simulating neural data $F(\kappa)_t$, which we assumed to correlate with $\kappa_t$, and included these data when estimating the model. Thus, the following comparison comprises three methods: LOTO, Bayesian modeling (without joint modeling), and the joint modeling approach.

We compared the three methods with respect to the correlation between the true values of $\kappa_t$ and the methods' estimates (*Figure 9—figure supplement 1A*) and with respect to the correlation between the neural data $F(\kappa)_t$ and the methods' estimates (*Figure 9—figure supplement 1B*). First of all, all three methods provided significant relationships with the true model parameters and with the neural data (all $p < .001$, all $BF_{10} > 1,000$). Expectedly, the correlations illustrate that the Bayesian modeling and the joint modeling approach yielded similar estimates. Averaged over the simulated participants, however, the joint modeling approach outperformed (non-joint) Bayesian modeling with respect to both $\kappa_t$ ($t(29) = 3.61$, $p = .001$, $d = 0.66$, $BF_{10} = 21.96$) and $F(\kappa)_t$ ($t(29) = 4.32$, $p < .001$, $d = 0.79$, $BF_{10} = 131.84$). Remarkably, LOTO outperformed Bayesian modeling ($t(29) = 2.82$, $p = .009$, $d = 0.51$, $BF_{10} = 6.07$) as well as the joint modeling approach ($t(29) = 2.51$, $p = .018$, $d = 0.46$, $BF_{10} = 3.33$) with respect to $\kappa_t$. With respect to the neural signal $F(\kappa)_t$, however, LOTO yielded similar correlations compared to Bayesian modeling ($t(29) = -0.62$, $p = .540$, $d = -0.11$, $BF_{10} = 0.26$) and was outperformed by the joint modeling approach ($t(29) = -3.03$, $p = .005$, $d = -0.55$, $BF_{10} = 13.78$). Overall, the joint modeling approach did indeed provide the best resultsin terms of linking the single-trial neural data to the cognitive model, but the differences between the methods appear to be comparatively small. Taking flexibility and difficulty of implementation into account, LOTO appears to offer an expedient alternative to joint modeling (see also 'Discussion').

## X. A power analysis for linking LOTO estimates to neural data

The ultimate goal of LOTO (and related approaches) is to establish a link between the inferred trial-by-trial variability in parameters of cognitive models and the brain activity measured with tools such as fMRI or EEG. Therefore, the question of whether LOTO is an appropriate tool for a given experiment with a specific design (including task, number of trials, number of participants etc.), cognitive model, and parameter(s) of interest should be answered by testing whether LOTO can be expected to recover the (truly existing) correlation between a neural signal and a parameter. In the case of modeling variability in more than one parameter, it is also important to test whether the potential misattribution of variability by LOTO may lead to incorrect inferences regarding the brain–parameter relationship (see 'Results: Sections V and VI'). We emphasize here that these requirements are not unique to LOTO. Any approach for investigating the neural correlates of trial-by-trial parameter variability should be accompanied by such analyses of statistical power (i.e., sensitivity) and false-positive rates (i.e., specificity).

To link LOTO estimates to neural data, we reverted to the HD-LBA model for which we have inferred variability in two parameters (i.e., discount factor $\kappa$ and decision threshold $b$) in 'Results: Section VI'. When simulating trial-specific parameter values $\kappa_t$ and $b_t$, as well as choices and RT, we also generated two hypothetical neural signals $F(\kappa)_t$ and $F(b)_t$ that were positively correlated with $\kappa_t$ and $b_t$, respectively (for details, see 'Materials and methods'). Then, we generated 1,000 datasets of 30 participants with 160 trials each and tested how often the LOTO estimate for a parameter (e.g.,

κ) correctly explains a significant amount of variability in the associated neural signal (e.g., $F[\kappa]$), and how often it incorrectly explains a significant amount of variability in the neural signal associated with the other parameter (e.g., $F[b]$) over and above that explained by the observations. This was realized by a random-effects linear regression analysis – similar to the General Linear Model approach in fMRI analysis (*Friston et al., 1994*). Stated differently, we tested i) the sensitivity of LOTO by means of a power analysis, asking how likely it is that LOTO identifies the correct neural signal, and ii) the specificity of LOTO by means of a false-positive analysis, asking how likely it is that LOTO misattributes a neural signal to a different parameter.

*Figure 10A* depicts the histograms of *p*-values for tests of whether LOTO's estimates for κ and *b* are significantly related to the brain signals $F(\kappa)_t$ (left panel) and $F(b)_t$ (right panel), using the setting with 30 participants. According to these results, LOTO achieves a statistical power (or sensitivity) of 67% for parameter κ and of 76% for parameter *b*, values that are slightly below the threshold of 80% that is usually deemed acceptable. On the other hand, the false-positive rate (or specificity) of LOTO is good, with values lying just slightly above the 5% error rate. This means that the small mis-attribution of parameter variability seen in 'Results: Section VI' does not carry over much to a misat-tribution of the neural signal. In *Figure 10B*, we repeated the analysis for a sample size that is increased by two-thirds (i.e., 50 instead of 30 simulated participants). Expectedly, the statistical power of LOTO improves and lies within acceptable ranges. An alternative solution is to increase the number of trials per participant. *Figure 10C* depicts the results when the number of participants is kept to 30 but the number of trials is increased by two-thirds (i.e., 267 instead of 160 trials). The improvement in statistical power is comparable to that resulting from the increased sample size. Accordingly, one would conclude from these analyses that investigating the neural basis of trial-by-trial variability in parameters κ and *b* of the HD-LBA model might require testing more than 30 par-ticipants or testing more than 160 trials per participant, whatever is more suitable in a given context (e.g., adding trials might be difficult for event-related fMRI designs that require long inter-trial inter-vals but less problematic for EEG designs).

## XI. An example of using LOTO for model-based fMRI

In this section, we apply LOTO to a real neuroimaging dataset to further illustrate the method's use and usability in model-based cognitive neuroscience. The dataset is taken from a previously pub-lished fMRI study (*Gluth et al., 2015*), in which participants were asked to make preferential choices from memory. More specifically, participants first encoded associations between food snacks and screen positions and then decided between pairs of food snacks while seeing only the snacks' screen positions (*Figure 11A*). Participants had either one or five seconds in which to retrieve the item–location associations and to deliberate upon the preferred snack option.

Choice behavior was modeled using an unbounded sequential sampling model that predicted more accurate decisions (i.e., decisions that are more consistent with evaluations of the food snacks in a separate task) with more deliberation time (details on the model and the fMRI analyses are pro-vided in 'Materials and methods'; see also Figure 3 in *Gluth et al. (2015)*). Importantly, the model estimates the average memory performance of a participant, that is, the probability of accurately remembering a snack when making decisions. In the original study (*Gluth et al., 2015*), we employed our previously proposed GR2017 method for estimating trial-by-trial variability in order to infer the likelihood of successful memory retrieval for each snack given the choice behavior (intui-tively, choosing unattractive snacks is attributed to poor memory performance). Here, we repeat this analysis but use LOTO to make the inference. Notably, LOTO takes advantage of the fact that each snack is a choice option in multiple (i.e., three) choice trials (see 'Results: Section II'). Ultimately, LOTO's item-specific estimate of memory performance is applied to the fMRI analysis of the encod-ing phase. In other words, a model-based analysis of the well-established *difference in subsequent memory effect* (*Paller and Wagner, 2002*) is conducted with the prediction that variations in mem-ory performance should be positively correlated with activation of the hippocampus (as has been found in our original publication).

Before using LOTO to inform the fMRI analysis, we checked whether the variability estimates could be linked to other behavioral indicators of memory performance and whether the assumption of systematic variability in memory could be justified. To address the first question, we conducted a logistic regression testing whether the LOTO-based estimate predicted correct responses in the cued recall phase (i.e., a phase following the decision-making phase in which participants were asked

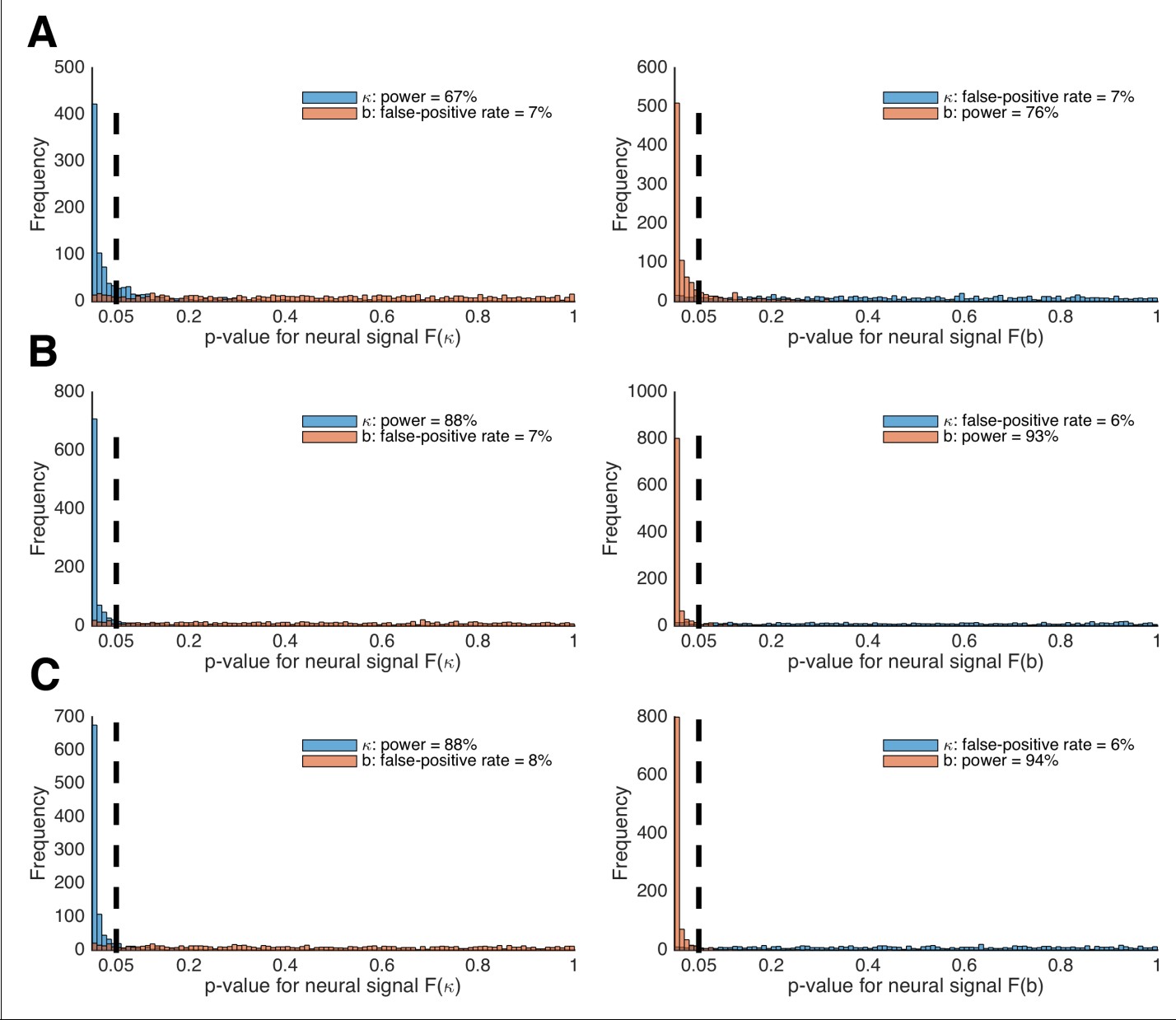

**Figure 10.** Power analysis of LOTO for detecting brain–parameter correlations. (**A**) Histogram of the *p*-values from tests of the significant relationships between κ and *b* and brain signals *F*(κ) (left panel) and *F*(*b*) (right panel). (**B**) The same as in panel **A** but for 50 instead of 30 participants per simulated experiment. (**C**) The same as in panel **A** but for 267 instead of 160 trials per participant.

DOI: https://doi.org/10.7554/eLife.42607.016

to recall each snack; see 'Materials and methods' and *Gluth et al., 2015*). As additional independent variables, this regression analysis also included whether the snack was encoded once or twice, how often a snack was chosen, and the subjective value of the snack. As can be seen in *Figure 12A*, the LOTO-based estimate of memory explained variance in cued recall performance over and above the other predictor variables ($t(29) = 7.44$, $p < .001$, $d = 1.36$, $BF_{10} > 1,000$). To address the question of systematic variability, we followed the procedures described in 'Results: Section V' and generated 1,000 datasets under the null hypothesis of no variability in memory performance. We then compared the LOTO-based improvement in model fit in these datasets with the improvement in the real data. Indeed, the improvement in the real data was much larger than in any of the datasets generated under the null hypothesis (*Figure 12B*). Furthermore, we checked whether

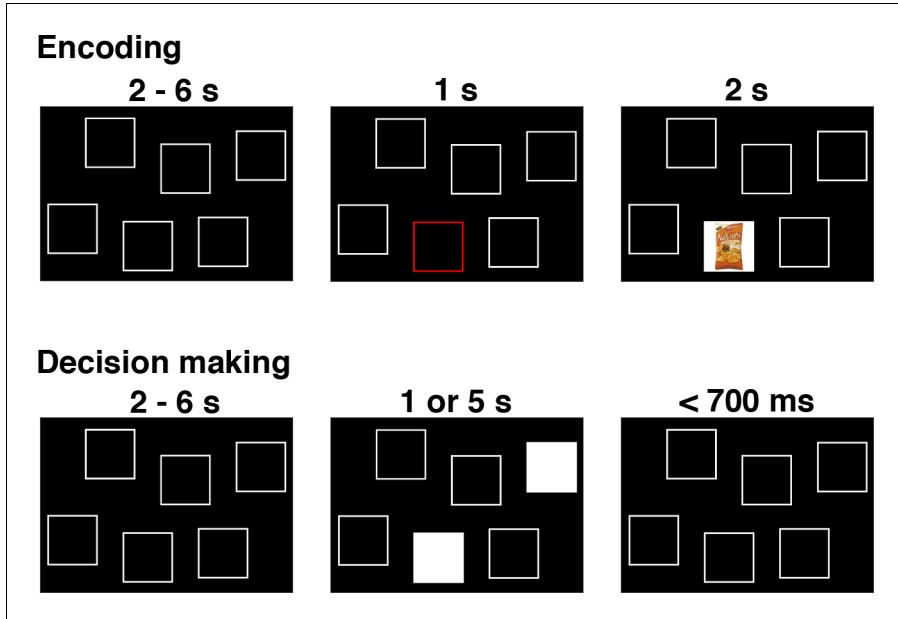

**Figure 11.** The remember-and-decide task (*Gluth et al., 2015*). This figure shows one example trial from the encoding and decision-making phases of the task. During an encoding trial, participants learned the association between one out of six different snacks with a specific screen position. During a decision-making trial, two screen positions were highlighted, and participants had to retrieve the two associated snacks from memory in order to choose their preferred option. Here, we apply LOTO to a decision-making model in order to infer memory-related activity during encoding. For a more detailed illustration of the study design, see Figure 1 in *Gluth et al. (2015)*.
DOI: https://doi.org/10.7554/eLife.42607.017

potential variability in parameter σ, which represents the standard deviation of the drift rate (i.e., noise in the decision process), could have been misattributed to the memory-related parameters. (It seems less plausible to assume that variability in the decision process, in contrast to variability in memory, is item-specific. Nevertheless, we included this control analysis for instructional purposes.) Thereto, we repeated the test for the presence of systematic variability with simulated data that included different levels of true variability in σ (for details, see 'Materials and methods'). As can be seen in *Figure 12—figure supplement 1*, LOTO does not appear to misattribute variability in σ to the memory-related parameters.

Having established that LOTO has the potential to identify meaningful neural correlates of item-specific memory performance, we turned to the fMRI analysis. Here, we employed a frequent approach to model-based fMRI (*O'Doherty et al., 2007*), which is to add the predictions of a cognitive model as a so-called parametric modulator (PM) of the hemodynamic response during a time window of interest. In our case, this time window is the period during which snack food items are presented during encoding, with the LOTO-based variability estimates serving as the PM (in order to test for a model-based subsequent memory effect). As can be seen in *Figure 12C*, this analysis revealed fMRI signals in bilateral hippocampus (MNI coordinates of peak voxel in left hippocampus: x = −18, y = −10, z = −12; right hippocampus: 38,–12, −26) that surpassed the statistical threshold corrected for multiple comparisons (left: $Z = 3.73$; $p = .033$; right: $Z = 4.46$, $p = .002$). These effects are in line with those reported in the original publication, which were based on the GR2017 method and also showed a significant relationship between item-specific memory strength and hippocampal activation. (Note that, as in the original publication, we ruled out alternative explanations for these results by including cued recall performance and subject values of snacks as additional PMs in the analysis; see 'Materials and methods'.)

To illustrate a second analysis approach, we also conducted a region-of-interest (ROI) analysis by extracting the average fMRI signal from anatomically defined masks of the left and right hippocampi (*Figure 12D*). The average signals in these ROIs were significantly related to the LOTO-based PM (left: $t(29) = 2.93$, $p = .007$, $d = 0.53$, $BF_{10} = 6.42$; right: $t(29) = 4.36$, $p < .001$, $d = 0.80$,

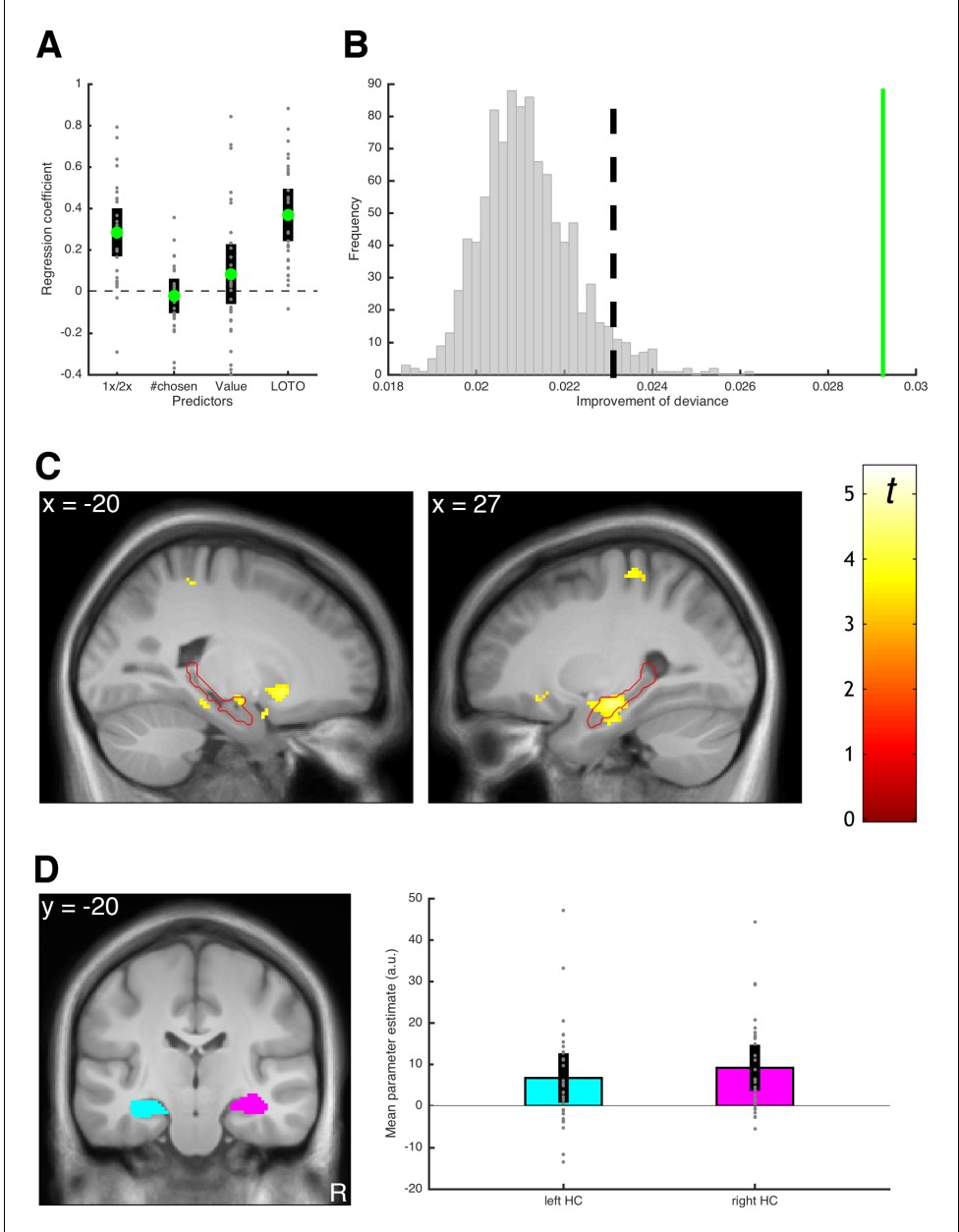

**Figure 12.** Results of the example LOTO application. (A) Average standardized regression coefficients for predicting memory performance in cued recall (1x/2x refers to how often an item was encoded). (B) Testing for systematic variability in memory; the green line shows the average trial-wise improvement of fit for the real dataset. The vertical dashed line represents the significance threshold based on the simulated data under the null hypothesis of no variability (gray histogram). (C) fMRI signals linked to the LOTO-based estimate of item-specific memory; the red outline depicts the anatomical mask of the hippocampus. (D) ROI analysis with anatomical masks of left and right hippocampus (HC) and average fMRI signals in these ROIs (the analyses in panels C and D are analogous to those in Figure 5 of *Gluth et al., 2015*). Individual values (gray dots) are shown together with 95% confidence intervals of the mean (black error bars).

DOI: https://doi.org/10.7554/eLife.42607.018

The following figure supplement is available for figure 12:

**Figure supplement 1.** Testing for potential misattribution of variability in 'decision noise' parameter σ.

DOI: https://doi.org/10.7554/eLife.42607.019

$BF_{10}$ = 182.52). Taken together, these findings demonstrate that we were able to reproduce the analysis of a model-based subsequent memory effect reported by *Gluth et al. (2015)* using LOTO.

## Discussion

In this work, we introduce LOTO, a simple yet efficient technique to capture trial-by-trial measures of parameters of cognitive models so that these measures can be related to neural or physiological data. The core principle is to infer single-trial values from the difference in the parameter estimates when fitting a model with all trials vs. a model with one trial being left out. The application of LOTO is simple: first, the full set of parameters of a model is estimated for all $n$ trials; then, the parameters of interest are estimated again $n$ times with each trial being left out once, while the remaining parameters are fixed to their all-trial estimates. In addition to its simplicity, LOTO is also comparatively fast: if fitting a model once to all trials takes $u$ time units, then LOTO's runtime will be $u \times n$ at most (in fact, it can be expected to be even faster because not all parameters are estimated again). And in contrast to the GR2017 approach, the LOTO runtime will not increase (much) if variability in more than one parameter is estimated. Compared to related methods, LOTO appears to provide mostly comparable or even superior results. Finally, it is a very general approach for capturing variability in parameters. In principle, it can be applied to any cognitive model, although we stress the importance of testing the feasibility of LOTO for each model and study design anew.

### The LOTO recipe

In this section, we provide a 'LOTO recipe' that summarizes the critical steps, tests and checks that one should follow when seeking to apply the method. We link each point to the relevant section(s) in the 'Results'. Thus, despite our recommendation that readers with limited statistical and mathematical knowledge should first consult this recipe, it is indispensable for interested cognitive neuroscientists to read the relevant sections before trying to apply LOTO to their own data. Notably, Steps 1 and 2 of this recipe are to be performed before conducting the experiment. Therefore, LOTO is not suitable for post-hoc analyses of data that have already been acquired with different study goals in mind. Also note that all of the computer codes associated with this article and with LOTO are freely available on the Open Science Framework (https://osf.io/du85f/).

1. Specify the cognitive model and the parameters for which trial-by-trial variability should be estimated; similarly, specify the design of the study (number of trials and participants, experimental manipulations, neuroscientific method).
   a. What type of observations does the model explain? If it explains only choices, try to extend the model to allow the prediction of RT ('Results: Section IV').
   b. How are the experimental manipulations or conditions related to the model (i.e., which manipulation is predicted to selectively influence a given parameter)? If the model has to be fit to each condition separately, try to adjust the model so that its predictions change directly as a function of the condition ('Results: Section III').
   c. Should variability be captured in each trial or across multiple trials (e.g., small blocks of trials)? If simulations of LOTO (see Step 2 below) do not yield acceptable results, consider pooling multiple trials into small blocks and capturing parameter variability from block to block ('Results: Section II').
   d. Does the neuroscientific method allow the measurement of signals that can be related to trial-by-trial variability? Block-designs for fMRI and ERPs for EEG are less suitable than event-related designs for fMRI and time-frequency analyses for EEG.
2. Test the feasibility of the LOTO approach for the chosen model and study design.
   a. Test whether LOTO can (in principle) provide sensible estimates of the true trial-specific values of the targeted model parameter(s). Specification of the first derivative of the log-likelihood and the Fisher information (as described in 'Results: Sections II and III') is desirable but often cumbersome or even impossible. Instead, we suggest running simulations ('Results: Sections II–IV').
   b. Check whether LOTO might misattribute random noise by conducting the non-parametric test for the presence of systematic variability ('Results: Section V'). Similarly, check whether LOTO might misattribute variability in other parameters to the parameters of interest by conducting the regression analyses outlined in 'Results: Section VI'.
   c. Estimate the statistical power and if applicable the specificity of LOTO given the model and study design ('Results: Section X'). This might require running a pilot study to obtain

a rough estimate of the amount of trial-by-trial variability in model parameters that can be expected (alternatively, existing datasets could be inspected for this purpose).

d. For experiments with many trials per participant (i.e., $\geq$1,000), check whether the default tolerance criterion of the parameter estimation method is sufficient or whether there are discrete 'jumps' in the LOTO estimates. If necessary, choose a stricter criterion ('Results: Section VII').

e. Always use the actually planned number of trials, number of participants and manipulations when doing simulations. Always include the observations that are used to estimate parameters (e.g., choices and RT) when testing the potential of LOTO to recover parameter variability. Applying LOTO is meaningless when the method does not yield any additional information over and above that provided by the observations themselves ('Results: Section II').

f. If the simulations (see Steps 2a,b) do not yield acceptable results, consider increasing the number of participants or the number of trials per participant ('Results: Section X'), or consider adapting the model or experimental paradigm ('Results: Sections II–IV').

3. After conducting the study, apply LOTO ('Results: Section I'), conduct the non-parametric test for the presence of systematic variability ('Results: Section V'), and use LOTO's estimate for analyzing the recorded neural data (e.g., 'Results: Section XI').

Of note, the test for the presence of variability ('Results: Section V'; Steps 2b and 3) and the power analyses ('Results: Section X'; Step 2c) are very time-consuming because they require the simulation of a large number (e.g., 1,000) of entire experiments and the application of LOTO to the resulting data. For example, for the first power analysis reported in 'Results: Section X', we ran 1,000 simulations of 30 participants with 160 trials each, meaning that the HD-LBA model had to be estimated ~5 million times. Therefore, it may not be possible to conduct an extensive search for optimal sample size and trial number to meet a desired statistical power such as 80%. Instead, trying out a limited set of realistic scenarios appears most appropriate. Note, however, that other approaches such as GR17 or hierarchical Bayesian modeling can be much more time-consuming than LOTO, so that even the simplest simulation-based power analyses would become infeasible using these approaches.

## The principle of leaving one out

The idea of leaving one data point out, also termed the 'jackknife' principle, is not new. In fact, the approach was proposed by statisticians in the 1950s (*Quenouille, 1956*; *Tukey, 1958*) as a way to estimate the variance and bias of estimators when dealing with small samples. From a technical point of view, LOTO bears obvious similarities to statistical methods such as (Bayesian) leave-one-out cross-validation, which is employed in classification-based analyses (*Bishop, 2006*), Bayesian model evaluation (*Vehtari et al., 2017*), or the leave-one-subject-out approach, which is used to circumvent the non-independence error in statistical analyses of fMRI data (*Esterman et al., 2010*). Notably, the jackknife principle has also been adopted for inferring trial-specific onset latencies of ERPs (*Miller et al., 1998*; *Stahl and Gibbons, 2004*) and for quantifying trial-specific functional connectivity between different brain regions (*Richter et al., 2015*). Thus, a combination of LOTO for inferring variability in cognitive model parameters with the same principle used for identifying trial-specific brain activation patterns could be a promising approach for future research in model-based cognitive neuroscience (e.g., *Benwell et al., 2018*).

## LOTO as a two-stage approach to linking behavioral and neural data

As mentioned in the introduction, LOTO is one of several approaches that have been put forward to link computational models of cognition and behavior with neural data. *Turner et al. (2017)* proposed a taxonomy of these approaches based on whether a method: i) uses neural data to constrain the (analysis of the) cognitive model, ii) uses the cognitive model to constrain the (analysis of the) neural data, or iii) uses both types of data to constrain each other in a reciprocal manner. Within this taxonomy, LOTO is best classified as belonging to the second class, that is, it infers trial-wise values of the cognitive model to then inform the analysis of neural data. In other words, LOTO is a 'two-stage approach' to model-based cognitive neuroscience, in which the modeling of behavior precedes the analysis of brain data (as exemplified in 'Results: Section XI'). This approach misses out on the advantage of reciprocity exhibited by the joint modeling approach (*Turner et al., 2015*).

However, LOTO's strength lies in its ease, speed and generality of application. Moreover, it retains the flexibility of the two-stage approach with respect to identifying the neural correlates of trial-by-trial parameters: it does not require an a priori hypothesis of which brain region or EEG component might be related to variability in a model parameter. Instead, the variability estimate captured by LOTO can simply be plugged into, for instance, a GLM-based fMRI whole-brain analysis (*Gluth and Rieskamp, 2017*; *O'Doherty et al., 2007*; see also *Figure 12C*). Notably, LOTO also retains the flexibility of testing different functional relationships between brain and behavior (e.g., linear, quadratic, exponential) and thus lends itself to exploring different linking functions for subsequent, confirmatory studies.

## Concluding remarks

Today, neuroimaging and psychophysiological techniques such as fMRI, EEG and eye-tracking allow us to shed light on the process of single actions. However, the development of cognitive models has only partially kept up with these advances because the central tenet of cognitive modeling is still the identification of invariant features of behavior by averaging over repeated observations. LOTO has the potential to fill this gap, in particular because it will not be restricted to a handful of specific models or to a handful of experts in the field who are highly proficient in computational modeling.

## Materials and methods

**Key resources table**

| Reagent type (species) or resource | Designation | Source or reference | Identifiers |
|---|---|---|---|
| Software | MATLAB | MathWorks | RRID:SCR_001622 |
| Software | R Project for Statistical Computing | https://cran.r-project.org/ | RRID:SCR_001905 |
| Software | Just Another Gibbs Sampler (JAGS) | http://mcmc-jags.sourceforge.net/ | - |

### Simulations of Bernoulli and binomial distributions

We drew $n = 300$ values of $\theta_t$ from a uniform distribution over a range of .2 (e.g., [.4, .6]), and then generated one Bernoulli event per trial $t$ using $\theta_t$ as the underlying probability. LOTO was then used to recover $\theta_t$ by estimating and subtracting $\theta_t - \theta_{\neg t}$, for every $t$ according to *Equation 2*. This was done for every range from [0, .2] to [.8, 1] in steps of .01 (i.e., 80 times) and repeated 1,000 times. Pearson product-moment correlations between $\theta_t$ and LOTO$_t$ were calculated and averaged over the 1,000 repetitions.

In a second step, we generalized to the binomial distribution by assuming that $\theta$ remains stable over a limited number of trials $m$ ($m < n$). The goal was then to recover $\theta_m$ with LOTO. We chose four different levels of $m$ (i.e., $L(m) = \{1, 5, 10, 20\}$), but kept $n$ constant (i.e., $n = 300$). Thus, the number of different values of $\theta_m$ decreased by $n/m$ as $m$ increased. Again, simulations were run for every range from [0, .2] to [.8, 1] in steps of .01 and repeated 1,000 times, before LOTO was applied (*Equation 5a*) and correlations between $\theta_m$ and LOTO$_m$ were calculated and averaged over the 1,000 repetitions (for each level of $m$ separately).

For 'Results: Section IX', which discusses the shape of the distribution of LOTO estimates, we conducted two simulations of $n = 10,000$ trials each, sampling 500 values of $\theta_m$ for blocks of 20 trials. The 500 values were drawn from a normal distribution (with mean = 0.5 and SD = 0.05) in the first simulation and from a uniform distribution (in the range [.4, .6]) in the second simulation. We then generated the corresponding Bernoulli events and applied LOTO.

### Simulations of intertemporal choice sets and the HD model

For the sake of plausibility, choice sets for the intertemporal choice task were created in rough correspondence to typical settings in fMRI studies (*Peters et al., 2012*). We chose to simulate 160 trials per 'participant', which would result in a realistic total duration of 40 min of an fMRI experiment, in which one trial lasts ~15 s (e.g., *Peters and Büchel, 2009*). As is often the case in fMRI studies on intertemporal choice, the immediate choice option was identical in all trials (i.e., $d_i = 0$; $x_i = 20$), but

the delay and the number of the delayed choice options varied from trial to trial. In a first step, delays $d_j$ were drawn uniformly from the set $S(d_j)$ = {1, 7, 14, 30, 90, 180, 270, 360}, and amounts $x_j$ were drawn from a uniform distribution with limits [20.5; 80], discretized to steps of 0.5. In a second step, this procedure was repeated until a choice set was found for which the utility of the immediate option $u_i$ was higher than the utility of the delayed option $u_j$ in ~50% of the trials, given the simulated participant's baseline parameter κ (see below). Specifically, a choice set was accepted when $u_i$ − $u_j$ > 0 in more than 45% but less than 55% of trials. Otherwise, a new choice set was created. This was repeated up to 1,000 times. If no choice set could be found, the criterion was relaxed by 10% (i. e., a choice set was now accepted when $u_i$ − $u_j$ >0 in more than 40% but less than 60% of trials). This procedure was repeated until an appropriate choice set was found. Note that using such adaptive designs is very common and recommended for intertemporal choice tasks in order to avoid having participants who choose only immediate or only delayed options (*Gluth et al., 2017*; *Koffarnus et al., 2017*; *Peters et al., 2012*). In practice, adaptive designs require a pre-test to infer the individual discount rate κ.

Simulations of the HD model were also based on empirical data (*Peters et al., 2012*). For each simulated participant, we first determined 'baseline' parameters κ and *β* (for the sake of simplicity, we omit the indicator variable for participants here and for all following notations) by drawing from log-normal distributions truncated to ranges [0.001, 0.07] and [0.01, 10], respectively. Means (μ) and standard deviations (σ) of these log-normal distributions were {μ_κ = –4.8, σ_κ = 1} and {μ_κ = –0.77, σ_κ = 0.71}, respectively, and were chosen so that the medians and interquartile ranges of the generated values would match those reported by *Peters et al. (2012)*. Log-normal distributions were used to account for the fact that distributions of empirical discount rates and sensitivity parameters are often right-skewed. If trial-by-trial variability in κ and *β* was assumed, then this was realized by drawing trial-specific values of $κ_t$ and $β_t$ from a normal distribution (truncated to values $\geq$ 0) with means κ and *β*, and standard deviations 0.01 and 1, respectively. Otherwise, we assumed that $κ_t$ = κ, $β_t = β$.

After specifying choice sets and trial-specific parameter values, choices were simulated and the parameters of the HD model were estimated for all trials. This was done in a two-step procedure starting with a grid search (with 25 different values per parameter) to find suitable starting values for the parameters, which were then used in a constrained simplex minimization. Then, LOTO was applied by leaving each trial out once and estimating the parameter(s) of interest (i.e., either $κ_t$ or $β_t$ or both), again using the 'all-trials' parameter estimates as starting values. Similarly, single-trial fitting was conducted whenever necessary by fitting each trial individually, also using the 'all-trials' estimates as starting values.

Unless stated otherwise, this procedure was conducted for 30 participants and repeated multiple times whenever necessary. The example participant shown in *Figure 3* was taken from such a set of simulations. This participant had baseline parameters κ = 0.0076 and *β* = 2.11, but only $κ_t$ varied from trial to trial. When testing for the presence of systematic trial-by-trial variability in 'Results: Section V', we generated 1,000 simulations of 30 participants under the null hypothesis and 10 simulations of 30 participants assuming 10, 20, 30, 40, 50, 60, 70, 80, 90, 100% variability in κ or *β* compared to the amount of variability used in our initial simulations (see above). When testing for potential misattribution of variability in amounts $x_j$, delays $d_j$, or utilities $u_j$ of the delayed option, we simulated 10 different levels of variability (i.e., Gaussian noise) in these input or output variables. Importantly, we ensured comparability with effects of κ by adjusting the largest amount of Gaussian noise in $x_j$ (i.e., $SD_x$ = 9), $d_j$ (i.e., $SD_d$ = 10), or $u_j$ (i.e., $SD_u$ = 6), so that the variance in the output variable $u_j$ would be similar to that under the largest amount of Gaussian noise in κ (i.e., $SD_κ$=0.01; see above). In the context of these simulations (see 'Results: Section V'), we also extended the intertemporal choice task to three options. For every trial of this task, three amounts and three delays were randomly drawn from the range of values used in the standard version of the task (see above) and were always combined so that one option had the smallest amount and delay, one option had the largest amount and delay, and one option had medium amount and delay. As for the standard version, trial generation was adapted to the sampled discount factor, so that choice probabilities would not be too extreme. Thereto, the choice probability of each option was calculated using the (generalized) logistic or soft-max choice function:

$$p_i = \frac{\exp(\beta * u_i)}{\sum_j \exp(\beta * u_j)}, \tag{10}$$

and amount and delays were sampled until choice probabilities were as similar as possible. (More specifically, the first set of options was accepted only if the highest average choice probability for one option was $\leq$ .34; for each subsequent set of options, this criterion was relaxed by an increase of .001 until a set of options was accepted.) *Equation 10* was also used to generate simulated choices, to estimate parameters, and to apply LOTO. Note that a three-option design could induce specific choice phenomena, so-called 'context effects', which are incompatible with the logistic or soft-max choice rule (*Busemeyer et al., 2019*; *Gluth et al., 2017*). For the purposes of the present work, however, these effects can be neglected.

## Simulations of the HD-LBA model

Procedures for simulating the HD-LBA model were similar to those for the HD model. Like the HD model, the HD-LBA model includes parameter $\kappa$, and values for $\kappa$ and $\kappa_t$ were generated in the same way as before. However, HD-LBA does not use the soft-max choice function (see *Equation 7*) to specify choice probabilities but instead uses the LBA model (*Brown and Heathcote, 2008*) to specify the joint choice and response time (RT) distributions. Thus, instead of including the soft-max choice sensitivity parameter $\beta$, the HD-LBA model includes parameter $\lambda$, which scales the relationship between utility and drift rate (see *Equations 9a and 9*b). HD-LBA also includes four additional parameters from the LBA model: parameter $s$, which models (unsystematic) trial-by-trial variability in the drift rates (i.e., drift rates for options $i$, $j$ are drawn from independent normal distributions with means $v_i$, $v_j$ and standard deviation $s$); parameter $b$, which represents the decision threshold; parameter $A$, which represents the height of the uninform distribution of the start point; and parameter $t_0$, which accounts for the non-decision time of the RT. Most of these parameters were kept identical for all simulated participants (i.e., $\lambda$ = 0.1, $s$ = 0.3, $A$ = 1, $t_0$ = 0.5). Besides $\kappa$, only the decision threshold $b$ varied across participants (because it was used to test how well LOTO captures trial-by-trial variability in two parameters simultaneously) by drawing individual values from a uniform distribution with limits [2, 3]. When testing LOTO on two parameters simultaneously, trial-specific values of $\kappa_t$ were drawn as stated above, and trial-specific values of $b_t$ were drawn from a normal distribution with mean $b$ and standard deviation 0.04 (with the constraint that $b_t \geq A$).

Parameter estimation again started with an initial grid search. To reduce computation time, grid search was extensive (i.e., 15 grid search values) for parameters $\kappa$ and $b$ but limited (i.e., three grid search values) for the remaining parameters that were identical across simulated participants. As for the HD model, the best parameter estimates from the grid search were used as starting values for the constrained simplex minimization, followed by applying LOTO and single-trial fitting.

For 'Results: Section IX', which discusses the shape of the distribution of LOTO estimates, the HD-LBA model was simulated with a few modifications. First of all, the number of trials was increased to 480 to improve the interpretability of the shape of distributions. Second, the parameters of the log-normal distribution from which individual values of $\kappa$ were drawn were changed to $\{\mu_\kappa$ = -3.5, $\sigma_\kappa$ = 0.2$\}$, so that single-trial parameter values were unlikely to be extreme (i.e., $\kappa_t$ = 0)–which would have affected the shape of the distribution. Normally distributed single-trial values of $\kappa_t$ and $b_t$ were drawn as specified above, whereas uniformly distributed values of $\kappa_t$ were drawn from a range [$\kappa - K/2$; $\kappa + K/2$] with $K$ = 0.35, and uniformly distributed values of $b_t$ were drawn from a range [$b - B/2$; $b + B/2$] with $B$ = 1.4.

## Simulations of the LBA model

We compared four different methods for capturing trial-by-trial variability with respect to how well these methods recover the trial-wise drift rate and the start point values of the LBA model (*Brown and Heathcote, 2008*). We chose these two LBA parameters because one of the tested methods, the single-trial LBA (STLBA) approach (*van Maanen et al., 2011*), has been developed specifically for this set of parameters. The other three methods were LOTO, the 'single-trial fitting' method (see 'Results: Section III'), and an approach that we proposed in a previous publication (*Gluth and Rieskamp, 2017*), 'GR17'.

In total, we generated synthetic data from 20 experiments with 20 simulated participants each, and with 200 trials per participant. The decision threshold and non-decision time parameters were fixed across participants (i.e., $b$ = 1.5; $t_0$ = 0.2). The remaining three parameters of the LBA model were sampled individually for each participant: i) a value representing the average drift rate parameter $\nu$ was drawn from a normal distribution with mean $\mu_\nu$ = 0.6 and standard deviation $\sigma_\nu$ = 0.02; ii) a value representing the standard deviation parameter $s$ (for trial-by-trial variability of the drift rate) was drawn from a normal distribution with $\mu_s$ = 0.2 and $\sigma_s$ = 0.02; and iii) a value representing the height of the start point distribution $A$ was drawn from a normal distribution with $\mu_A$ = 1 and $\sigma_A$ = 0.1 (with the constraint that $A \leq b$). Trial-specific drift rates $\nu_t$ were drawn from a normal distribution with $\mu(\nu_t) = \nu$ and $\sigma(\nu_t) = s$. Trial-specific start points $A_t$ were drawn from a uniform distribution with limits [0, $A$]. Note that these values refer to the (trial-specific) drift rates and start points for correct responses (as, for instance, in a two-alternative forced-choice perceptual or inferential choice task). Similarly, trial-specific drift rates and start points for incorrect responses were drawn from distributions with the same specifications, except that the mean of the drift rate was $1-\nu$. We chose the above-mentioned values so that accuracy rates and RTs for simulated participants were kept in realistic ranges (i.e., mean accuracy rates and RTs per simulated participant ranged from 53 to 89% and from 1.5 to 2.1 s, respectively).

Parameter estimation procedures were very similar to those of the HD-LBA model. An initial grid search with an extensive search (i.e., 15 grid search values) for parameters $\nu$ and $A$ and a limited search (i.e., three grid search values) for the remaining parameters was used to find good starting values for the constrained simplex minimization, followed by the application of the four methods for capturing trial-by-trial variability. The STLBA method was applied according to the equations provided by *van Maanen et al. (2011)*. The GR17 method required the specification of a range of possible parameter values for $\nu$ and $A$ as well as a step size with which this range of values was scanned (i.e., prior, likelihood and posterior values were calculated for each step within the range of possible values; for more details, see *Gluth and Rieskamp, 2017*). Possible values for both parameters were restricted to be within three standard deviations below and above the parameters' group means, with 1,000 steps per parameter. This implied 1 million calculations for both prior and likelihood distributions because the joint distributions of both parameters needed to be specified. (With these specifications, the GR17 method required about three times the computation time of the three other methods combined; this extensive computation time was the reason for reducing the number of participants in these simulations from 30 to 20.) To test the methods' abilities to recover trial-specific drift rates and start points, we restricted the analysis to correct trials only (as is often the case in perceptual or inferential choice tasks).

## Bayesian modeling and joint modeling

For the second comparison of LOTO with related approaches in 'Results: Section IX', Bayesian modeling (*Farrell and Lewandowsky, 2018*; *Lee and Wagenmakers, 2013*) and the joint modeling approach (*Palestro et al., 2018*; *Turner et al., 2015*) were implemented for the HD model using the Just Another Gibbs Sampler (JAGS) software. Our specifications of the joint modeling approach closely followed the tutorial by *Palestro et al. (2018)*: the 'behavioral' parameter $\delta$ and the 'neural' parameter $\theta$ were drawn from a joint, multivariate normal distribution with means $\mu_\delta$ and $\mu_\theta$ and precision matrix $\Omega$. The means $\mu_\delta$ and $\mu_\theta$ were drawn from a multivariate normal distribution with means [0, 0] and precision matrix [1, 0; 0, 1], and $\Omega$ was drawn from a Wishart distribution with dispersion matrix [1, 0; 0, 1] and df = 2. The neural data were modeled as being sampled from a normal distribution with mean $\theta$ and standard deviation $\tau$ = 2 (the simulation of neural data is explained in the next paragraph). The behavioral data were sampled from the HD model. Within this behavioral model, the discount factor $\kappa$ was the probit-transformed value of $\delta$ (because $\kappa$ is restricted to values between 0 and 1, but $\delta$ is drawn from a normal distribution). To reduce the joint modeling approach to the (non-joint) Bayesian modeling, the sampling of neural data was simply omitted, so that parameter estimates were solely dependent on the behavioral data. For the sake of simplicity, we conducted parameter estimation within each simulated participant and did not draw individual parameters from higher-level group distributions. Sampling settings were as follows: five chains, 20,000 burn-in samples per chain, 2,000 recorded samples, thinning of 20 samples. Model

convergence was checked by computing the Gelman-Rubin statistic $\hat{R}$ (*Gelman and Rubin, 1992*), which was ensured to be below 1.01.

## Simulations of neural signals

To apply the joint modeling approach (*Turner et al., 2015*) and to test whether LOTO can capture neural signals that systematically vary with variability in a cognitive model parameter, we simulated neural signals in a very basic form. We assumed that the neural signal $F(\pi)$ in trial $t$ (which could, for instance, represent the amplitude of the fMRI blood-oxygen-level-dependent signal in a circumscribed region of the brain or the average EEG spectral power in a specific frequency band during a circumscribed time period of the trial) that is systematically linked to variability in a parameter $\pi$ is drawn from a normal distribution with mean $\pi_t$ and standard deviation $\sigma_{F(\pi)}$ (see *Palestro et al., 2018* for a similar assumption of normally distributed neural data in related work). The value for the standard deviation $\sigma_{F(\pi)}$ was specified as follows: we simulated $F(\pi)_t$ together with $\pi_t$ and the observations (choices and RT) for the desired number of trials (160) and participants (30), and then conducted linear regressions with $F(\pi)_t$ as the dependent variable and with $\pi_t$ and the observations as independent variables for each simulated participant (independent variables were standardized). Then, we estimated the effect size of the regression coefficient of $\pi_t$ at the group level. The goal was to obtain an effect size of $d = 1$ by trying out different values for $\sigma_{F(\pi)}$ (the choice of the effect size is justified in the next paragraph). This iterative procedure was stopped as soon as a value of $\sigma_{F(\pi)}$ was found, for which the 95% confidence interval of the effect size overlapped with 1 when generating 1,000 simulations of the desired number of participants and trials (i.e., 20/30 participants with 200/160 trials in 'Results: Sections IX and X'). For parameter $\kappa$ of the HD model, we obtained $\sigma_{F(\kappa)} = 7.5$. For parameter $\kappa$ of the HD-LBA model, we obtained $\sigma_{F(\kappa)} = 0.93$; for parameter $b$ of the HD-LBA model, we obtained $\sigma_{F(b)} = 4.8$. With respect to the HD-LBA model, this implies lower variability in the simulated neural signal for parameter $\kappa$ than in the signal for parameter $b$, because the trial-by-trial variability in the former parameter is much lower than that in the latter (see above). This might be implausible from a neurophysiological point of view. However, this implausibility does not matter in the context of the current work because we are only interested in the correlation between $F(\pi)_t$ and $\pi_t$ (and the LOTO estimate associated with $\pi_t$), which is unaffected by a linear transformation of $F(\pi)_t$.

The rationale of this approach is that in an idealized world, in which we would know the true trial-specific values of $\pi_t$, we could expect a very high effect size (such as $d = 1$) for detecting a relationship between $\pi_t$ and $F(\pi)_t$. This effect size is slightly higher than, for instance, current estimates of realistic effect sizes for fMRI studies, which are based on large-scale empirical studies (*Geuter et al., 2018*; *Poldrack et al., 2017*). But again, the chosen effect size refers to a hypothetical world, in which the true trial-by-trial value of a cognitive function is known and then related to brain activity. For such a fortunate scenario, we can expect to obtain a slightly higher effect size than in reality.

We generated 1,000 datasets of 30 simulated participants with 160 trials for the HD-LBA model. For each of these datasets, we conducted random-effects linear regression analyses, that is, we ran two linear regression analyses with $F(\kappa)_t/F(b)_t$ as the dependent variable and the LOTO estimates for $\pi_t$ and $b_t$ together with the observations (i.e., choices and RT) as independent variables in each simulated participant, and then conducted two-sided one-sample $t$-tests on the regression coefficients for $\pi_t$ and $b_t$ at the group level. The proportion of significant effects for 'correct'/'wrong' pairings of $F$ and $\pi$ determine the statistical power/false positive rate of LOTO.

## A note on the statistical analyses for simulations

The employment of frequentist statistical analyses for simulations is controversial because the results depend on the degrees of freedom of the tests, which can easily be adjusted to change the significance of a result. Our motivation in providing these statistical tests is that they offer an intuition of whether specific settings or features can be expected to be relevant when applying LOTO 'in the field'. Therefore, we chose to conduct statistical tests with their degrees of freedom usually being in the range of common and realistic sample sizes in cognitive neuroscience (e.g., studies of 20 or 30 participants). To complement the frequentist statistics, we also report effect sizes (i.e., Cohen's $d$, correlation coefficients) and the results of Bayesian hypothesis tests (i.e., the Bayes Factor $BF_{10}$ refers

to evidence in favor of the alternative hypothesis). Bayesian hypothesis tests were conducted in R using the BayesFactor package.

## fMRI example: essential study information

Three groups with a total of 84 participants (*Gluth et al., 2015*) took part in the study (one group of participants that underwent fMRI and two groups that conducted behavioral tasks only). The final sample of the fMRI group included 30 participants, and the analyses in the current article are restricted to this sample. All participants gave written informed consent and the study was approved by the Ärztekammer Hamburg, a local ethics committee in Hamburg, Germany (case # PV4290), where the study was conducted.

Participants were asked to not to eat for 3 hr prior to the experiment. They were familiarized with a set of 48 common food snacks from German supermarkets. To infer the subjective values of food snacks, participants conducted a Becker-DeGroot-Marschak (BDM) auction (*Becker et al., 1964*), in which they stated their willingness to pay money for each snack. To incentivize accurate bids, one randomly selected bid trial was realized at the end of the experiment. Inside the MR scanner, participants conducted the remember-and-decide task, which consisted of multiple runs of encoding, distraction, decision-making and recall phases. During encoding, participants learned associations between six different food snacks and six different screen positions. To manipulate memory performance, half of the snacks were encoded once, and the other half twice. During distraction, participants worked on a 2-back working memory task for 30 s. Afterwards, participants made nine decisions between pairs of snacks. Snacks were not shown directly, but the respective screen positions were highlighted and participants had to use their memory to make accurate decisions. To manipulate deliberation time, participants were given either a 1 s or a 5 s time window before they had to make a choice. Finally, the ability of the participant to remember snacks correctly was assessed via cued recall (during this phase, MR scanning was paused to enable communication between the participant and the experimenter). Participants were incentivized to make accurate choices by realizing one choice from a randomly selected decision-making trial. Further information can be found in *Gluth et al. (2015)*.

## fMRI example: cognitive model

To predict preferential choices and to inform the LOTO-based fMRI analysis, we used a slightly modified version of the cognitive model proposed in the original study (each modification is explained below). This model assumes that decisions are based on a noisy evidence accumulation (diffusion) process that is informed by the subjective values of the available choice options. The model also accounts for the possibility that options are not recalled and for participants' tendency to prefer remembered over forgotten options (the *memory bias*; see also *Gluth et al., 2015* and *Mechera-Ostrovsky and Gluth, 2018*). Because participants were forced to wait for a specified amount of time (i.e., either 1 s or 5 s) and then had to indicate their decision promptly, the model does not assume a decision threshold but an unbounded diffusion process that is terminated at the specified amount of time. The probability that an option is chosen is thus given by the probability that the diffusion process ends with a positive or negative value. More specifically, the probability of choosing the left option (*l*) over the right option (*r*) is:

$$p(l|\{l,r\}) = \Phi\left(\frac{\Delta_V * \sqrt{t}}{\sigma}\right), \tag{11}$$

where $\Phi(x)$ refers to the value of the cumulative distribution function of the standard normal distribution at $x$, $t$ refers to the deliberation time (i.e., $t = 1$ or $t = 5$), $\sigma$ accounts for the amount of noise in the diffusion process ($\sigma \geq 0$), and $\Delta_V$ refers to the value difference between $l$ and $r$, which is given by:

$$\Delta_V = M_l * (V_l - g) - M_r * (V_r - g), \tag{12}$$

where $M_l$ is the probability that option $l$ is remembered, $V_r$ is the standardized subjective value of option $r$, and $g$ represents the memory bias. The memory probability $M$ is either $\alpha$ or $\beta$, depending on whether an item has been shown once or twice during encoding, and $\alpha$ and $\beta$ are two free parameters ($0 \leq \alpha, \beta \leq 1$). This definition differs from the model used in *Gluth et al. (2015)* in which

β represented the improvement of memory performance for items encoded twice compared to items encoded once. We changed this definition because here we are not interested in testing whether there is a significant improvement in memory for items encoded twice. The second adaption of the model is that the memory bias $g$ is not a free parameter. Instead, we took each participant's intercept coefficient from the logistic regression that tested for the presence of a memory bias (see Equation 1 in *Gluth et al., 2015*). This adaption was made because, contrary to the original study, we did not estimate parameters with hierarchical Bayesian modeling here, and because treating the memory bias as a free parameter led to severe misspecifications of the other free parameters in some participants when using maximum likelihood estimation.

The three free parameters of the model (i.e., σ, α and β) were estimated via maximum likelihood estimation using the same minimization techniques as were used for the simulations. A grid search was not performed, but the minimization was repeated ten times with randomly generated starting values. After estimating the full model, LOTO was applied to extract trial-by-trial (or more exactly, item-by-item) variability in memory probability $M$ by fixing σ to the full-model estimate and re-estimating either α or β with each item taken out once (e.g., for an item $i$ that was encoded twice, α was also fixed to the full-model estimate and β was re-estimated by taking out the three decision-making trials that included $i$ as a choice option). The difference between the $n$ item and the $n$–1 item parameter estimates were then computed and used for the fMRI analysis. To test for the presence of systematic variability in memory probability (see 'Results: Section V'), we generated 1,000 datasets of each participant using the full-model parameter estimates and assuming no variability in α or β. LOTO was then applied to these datasets and the improvement in model fit was compared to that in the real data. To check whether potential trial-by-trial variability in parameter σ could have been misattributed to the memory-related parameters, we simulated data with true trial-wise Gaussian noise in σ (but not in α or β) and compared the LOTO-based improvement in model fit with that of the distribution under the null hypothesis of no variability. We tried 10 different levels of noise in σ; for the highest level, the trial-by-trial variability within a simulated participant was as high as the across-subject variability in the real data (i.e., $SD_\sigma = 2.2$). The lower nine levels exhibited 10% to 90% of this variability.

## fMRI example: analysis of fMRI data

Because the fMRI analysis closely followed the procedures described in *Gluth et al. (2015)*, we provide only a brief overview here. Details on fMRI data acquisition, pre-processing and statistical analyses can be found in the original publication. Statistical analysis was based on the General Linear Model (GLM) approach as implemented in SPM12. As in the original study, twelve onset vectors were defined, including two vectors for the encoding phase: one for items shown once and one for items shown twice. Both of these two onset vectors were accompanied by three parametric modulators (PM) that modeled the predicted change of the hemodynamic response as a function of the following variables of interest: i) the subjective value of the encoded snack; ii) whether the snack was correctly remembered or not in the subsequent cued recall phase; and iii) the LOTO-based estimate of item-by-item variability in memory probability (see above). Importantly, the LOTO-based PM was entered into the GLM last, so that correlated fMRI signals could be uniquely attributed to this PM. For each participant, one contrast image was generated that combined the effect of the LOTO-based PM for both onset vectors of the encoding phase.

At the group level, individual contrast images were subjected to a one-sample $t$-test. For illustration purposes, we performed two different statistical analyses. In the first analysis, we tested for significant voxels within the bilateral hippocampus, using a family-wise error correction for small volume at $p < .05$. The small volume was defined by a bilateral anatomical mask of the hippocampus taken from the AAL brain atlas (*Tzourio-Mazoyer et al., 2002*). The results were displayed on the mean anatomical (T1) image of all participants using an uncorrected threshold of $p < .001$ with at least 10 contiguous voxels. The second analysis was a region-of-interest (ROI) analysis for which we averaged the fMRI signal for left and right hippocampus separately (using the same anatomical mask) and tested whether the average signal in the ROI was significant (at $p < .05$, two-sided).

## General settings

All simulations and analyses were conducted in MATLAB. Function minimization via a (constrained) simplex algorithm was conducted using the function *fminsearchcon*. For determining the convergence of parameter estimation and the termination of search, a very strict tolerance criterion of $10^{-13}$ was chosen, because LOTO needs to work with very small differences in deviance when trying to adjust a parameter value from the 'all-trial' estimate to the *n*–1 estimate, in particular when *n* is high. To illustrate the effect of the tolerance criterion, in 'Results: Section VII' the performance of LOTO for the HD model was compared between the strict tolerance criterion and the default criterion in MATLAB, which is $10^{-4}$. Bayesian and joint modeling were conducted with JAGS (see above), which was called from within MATLAB via the function matjags.m.

## Code availability

The data and computer codes associated with this article and with the LOTO method are freely available on the Open Science Framework (https://osf.io/du85f/). Although running these codes requires access to the proprietary MATLAB software, we stress that LOTO is a technique rather than a software package and can therefore be realized in various programing languages (including open source software such as R or Python).

## Derivatives of the hyperbolic discounting (HD) model

The utilities $u_i$ and $u_j$ of the immediate and delayed choice options, as well as the corresponding choice probabilities $p_i$ and $p_j$, according to the HD model are provided in *Equations 6 and 7*, respectively. The log-likelihood for a single choice of the immediate option is given by the natural logarithm of the choice probability $p_i$ and thus:

$$
\begin{aligned}
LL_i = \ln(p_i) \quad &= \ln\left[\frac{1}{1+\exp(-\beta*[u_i-u_j])}\right]\\
&= \ln[1] - \ln\left[1+\exp(-\beta*[u_i-u_j])\right]\\
&= -\ln\left[1+\exp(-\beta*[u_i-u_j])\right]
\end{aligned}
\tag{13}
$$

The derivative of $LL_i$ with respect to κ (which is hidden in $u_j$ in *Equation 13*) is therefore:

$$
\begin{aligned}
\frac{\partial LL_i}{\partial \kappa} &= -\frac{1}{1+\exp(-\beta*[u_i-u_j])} * \frac{\partial\left[1+\exp(-\beta*[u_i-u_j])\right]}{\partial \kappa}\\
&= -\frac{\exp(-\beta*[u_i-u_j])}{1+\exp(-\beta*[u_i-u_j])} * \frac{\partial(-\beta*[u_i-u_j])}{\partial \kappa}\\
&= -\left[1 - \frac{1}{1+\exp(-\beta*[u_i-u_j])}\right] * \frac{\partial\left(-\beta*\left[x_i-\frac{x_j}{1+\kappa*d_j}\right]\right)}{\partial \kappa} .\\
&= -(1-p_i) * \frac{\partial\left(\beta*x_j*[1+\kappa*d_j]^{-1}-\beta*x_i\right)}{\partial \kappa}\\
&= -(1-p_i) * -\left(\beta*x_j*[1+\kappa*d_j]^{-2}\right)\\
&= \frac{\beta*x_j*d_j}{[1+\kappa*d_j]^2} * (1-p_i)
\end{aligned}
\tag{14}
$$

Similarly, the log-likelihood for a single choice of the delayed option is:

$$
\begin{aligned}
LL_j = \ln(p_j) \quad &= \ln\left[\frac{1}{1+\exp(-\beta*[u_j-u_i])}\right]\\
&= \ln[1] - \ln\left[1+\exp(-\beta*[u_j-u_i])\right] .\\
&= -\ln\left[1+\exp(-\beta*[u_j-u_i])\right]
\end{aligned}
\tag{15}
$$

And the derivative of $LL_j$ with respect to κ is therefore:

$$
\begin{aligned}
\frac{\partial LL_j}{\partial \kappa} &= -\frac{1}{1+\exp\left(-\beta*\left[u_j-u_i\right]\right)} * \frac{\partial\left[1+\exp\left(-\beta*\left[u_j-u_i\right]\right)\right]}{\partial \kappa} \\
&= -\frac{\exp\left(-\beta*\left[u_j-u_i\right]\right)}{1+\exp\left(-\beta*\left[u_j-u_i\right]\right)} * \frac{\partial\left(-\beta*\left[u_j-u_i\right]\right)}{\partial \kappa} \\
&= -\left[1-\frac{1}{1+\exp\left(-\beta*\left[u_j-u_i\right]\right)}\right] * \frac{\partial\left(-\beta*\left[\frac{x_j}{1+\kappa*d_j}-x_i\right]\right)}{\partial \kappa} \\
&= -\left(1-p_j\right) * \frac{\partial\left(\beta*x_i-\beta*x_j*\left[1+\kappa*d_j\right]^{-1}\right)}{\partial \kappa} \\
&= -p_i * \left(\beta*x_j*\left[1+\kappa*d_j\right]^{-2}\right) \\
&= -\frac{\beta*x_j*d_j}{\left[1+\kappa*d_j\right]^2}*p_i
\end{aligned}
\tag{16}
$$

## Example derivatives of LOTO for the HD model

Here, we show that LOTO will produce estimates of a parameter that are related to both the choices and task features and that are not extreme, as soon as each option has been chosen at least two times (in the case of a two-alternative choice task). We will use the examples of the HD model mentioned in 'Results: Section III'. Assume that there are two trials with identical features (i.e., identical amount $x_i$ of the immediate option, identical amount $x_j$ of the delayed option, identical delay $d_j$ of the delayed option) and that the immediate option is chosen in one trial but the delayed option is chosen in the other trial. (We assume identical features only to simplify the derivatives. As stated above, it is important to vary the features across trials.) In this case, the derivative of the log-likelihood with respect to the discount factor $\kappa$ is given by *Equation 8c*. Setting this equation to 0 yields the MLE of the parameter:

$$
0=\frac{\partial LL_{\{i,j\}}}{\partial \kappa}=\frac{\partial LL_i}{\partial \kappa}+\frac{\partial LL_j}{\partial \kappa}=\frac{\beta*x_j*d_j}{\left[1+\kappa*d_j\right]^2}*(1-p_i)-\frac{\beta*x_j*d_j}{\left[1+\kappa*d_j\right]^2}*p_i.
\tag{17}
$$

We can easily generalize this to the case of $n$ trials in which the immediate option was chosen $n_i$ times and the delayed option was chosen $n_j$ times (i.e., $n_i + n_j = n$):

$$
0=n_i*\frac{\partial LL_i}{\partial \kappa}+n_j*\frac{\partial LL_j}{\partial \kappa}=n_i*\frac{\beta*x_j*d_j}{\left[1+\kappa*d_j\right]^2}*(1-p_i)-n_j*\frac{\beta*x_j*d_j}{\left[1+\kappa*d_j\right]^2}*p_i.
\tag{18a}
$$

This equation has to be solved for $\kappa$, which requires some steps. By putting the second part of the right-hand side to the left hand side, we get:

$$
n_i*\frac{\beta*x_j*d_j}{\left[1+\kappa*d_j\right]^2}*(1-p_i)=n_j*\frac{\beta*x_j*d_j}{\left[1+\kappa*d_j\right]^2}*p_i
$$

This simplifies to:

$$
n_i*(1-p_i)=n_j*p_i
$$

We re-arrange this to:

$$
\frac{n_j}{n_i}=\frac{(1-p_i)}{p_i}=\frac{1}{p_i}-1
$$

Note that $p_i$ is specified in *Equation 7*. Therefore:

$$
\frac{n_j}{n_i}=\frac{1}{\frac{1}{1+\exp\left(-\beta*\left[u_i-u_j\right]\right)}}-1=1+\exp\left(-\beta*\left[u_i-u_j\right]\right)-1=\exp\left(-\beta*\left[u_i-u_j\right]\right)
$$

Taking the logarithm on both sides yields:

$$
\ln\left(\frac{n_j}{n_i}\right)=-\beta*\left(u_i-u_j\right)=\beta*u_j-\beta*u_i
$$

We re-arrange this to:

$$\ln\left(\frac{n_j}{n_i}\right) + \beta * u_i = \beta * u_j$$

Note that $u_i$ and $u_j$ are given by **Equation 6**. Therefore:

$$\ln\left(\frac{n_j}{n_i}\right) + \beta * x_i = \beta * \frac{x_j}{1 + \kappa * d_j}$$

Finally, we re-arrange this to isolate κ:

$$\left(1 + \kappa * d_j\right) * \left(\ln\left[\frac{n_j}{n_i}\right] + \beta * x_i\right) = \beta * x_j$$

$$1 + \kappa * d_j = \frac{\beta * x_j}{\left(\ln\left[\frac{n_j}{n_i}\right] + \beta * x_i\right)}$$

$$\kappa * d_j = \frac{\beta * x_j}{\left(\beta * x_i + \ln\left[\frac{n_j}{n_i}\right]\right)} - 1$$

$$\kappa = \frac{1}{d_j} * \left[\frac{\beta * x_j}{\left(\beta * x_i + \ln[n_j] - \ln[n_i]\right)} - 1\right] \tag{18b}$$

For instance, in the case of $n_i = n_j$, this equation simplifies to:

$$\kappa = \frac{x_j - x_i}{d_j * x_i}$$

which yields a κ of 0.075 and 0.182 for the two example trials mentioned in 'Results: Section III' (compare with the right panels of **Figure 2**).

Importantly, Equation 18b can be expected to yield reasonable results as long as $n_i$ and $n_j$ are $\geq$ 1. If either $n_i$ or $n_j$ is 0, then the first term in the parentheses on the right-hand side of Equation 18b will be 0, because we have $\ln[0] = \pm\infty$ in the denominator, and the estimation of κ will simply be $-1/d_j$, which is negative and thus lies outside of the range of values allowed for κ. When applying LOTO, we therefore require $n_i$ and $n_j$ to be $\geq$ 2, because the parameter is estimated once for $n$ and once for $n$–1 trials, and in the $n$–1 case, LOTO will yield an unreasonable estimate in one trial (i.e., if $n_i = 1$, then the error will be in that one trial in which the immediate option was chosen; if $n_j = 1$, then the error will be in that one trial in which the delayed option was chosen).

## Acknowledgements

We thank the members of the Decision Neuroscience and Economic Psychology groups at the University of Basel for helpful discussions, and Gilad Zlotkin, Yoav Kessler and Gideon Rosenthal for suggesting related work in other domains. This work was supported by a grant from the Swiss National Science Foundation (#100014_172761) to SG and by a grant from the Israel Science Foundation (#381–15) to NM.

## Additional information

### Funding

| Funder | Grant reference number | Author |
| --- | --- | --- |
| Swiss National Science Foundation | 100014_172761 | Sebastian Gluth |
| Israel Science Foundation | 381-15 | Nachshon Meiran |

The funders had no role in study design, data collection and interpretation, or the decision to submit the work for publication.

## Author contributions
Sebastian Gluth, Conceptualization, Software, Formal analysis, Visualization, Methodology, Writing—original draft; Nachshon Meiran, Conceptualization, Methodology, Writing—review and editing

## Author ORCIDs
Sebastian Gluth (ID) http://orcid.org/0000-0003-2241-5103

## Ethics
Human subjects: All participants gave written informed consent, and the study was approved by the Aerztekammer Hamburg, Germany (case # PV4290). All experiments were performed in accordance with the relevant guidelines and regulations.

## Decision letter and Author response
Decision letter https://doi.org/10.7554/eLife.42607.024
Author response https://doi.org/10.7554/eLife.42607.025

# Additional files

## Supplementary files
• Transparent reporting form
DOI: https://doi.org/10.7554/eLife.42607.020

## Data availability
The relevant data and computer codes are uploaded on the Open Science Framework (https://osf.io/du85f/) and are freely available.

The following dataset was generated:

| Author(s) | Year | Dataset title | Dataset URL | Database and Identifier |
|---|---|---|---|---|
| Gluth S, Meiran N | 2018 | Leave-one-trial-out (LOTO) | https://osf.io/du85f/ | Open Science Framework, 10.17605/OSF.IO/DU85F |

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
