## [Decision Letter]

Thank you for submitting your article "Leave-One-Trial-Out (LOTO): A general approach to link single-trial parameters of cognitive models to neural data" for consideration by *eLife*. Your article has been reviewed by Michael Frank as the Senior Editor, a Reviewing Editor, and two reviewers.

The reviewers have discussed the reviews with one another and the Reviewing Editor has drafted this decision to help you prepare a revised submission.

Both of the reviewers appreciate the importance of readily available methods to estimate single-trial values of computational model parameters. We feel that there is much to like about the current approach, but there are some concerns, that I have compressed into a single list of points below.

Summary:

The authors discuss a leave-one-out method to estimate trial-by-trial variability in parameters of computational models (LOTO). The rationale is that the deviation between a model parameter that is estimated using all data and a model parameter that is estimated using all data minus one data point, reflects the influence of that one data point. The authors demonstrate that LOTO is relatively easy to implement and faster with respect to previous approaches, and importantly it can be easily demonstrated via simulations whether the adopted approach is falsifiable.

Essential revisions:

1) An important issue with the current version of the paper is that in subsection “Comparison of LOTO with related approaches”, a comparison with the more complex, but presumably superior method of joint modeling (Turner et al.) is not performed. It would be good to see how much information about the single trial parameter estimates is really lost when not accounting for the covariance structure as Turner proposes.

2) A substantial concern about the utility of the approach and the interpretation of the results obtained by LOTO comes from the fact that recent evidence indicates that a good proportion of trial to trial fluctuations comes from inference processes of the incoming signals or subsequent computations (for emerging evidence see e.g., Drugowitsch et al., 2016; Findling et al., 2018; Polania et al., 2018). This suggests, that for instance, in the case of the HD example given by the authors, fluctuations can come directly from encoding d_i_ or x_i_ or even from performing the computation (x_i_ /(1+κ*d_i_)) but not necessarily from κ.. This means that variability in behavior can be misattributed to a given parameter (and therefore misattributed to a brain region when applied to neural data), but the sources of fluctuations may potentially come from encoding/decoding processes of input-signals/computations as indicated by recent research. Is there a way that the authors can rule out or study this possibility? For the case of the HD model, I assume that this might be difficult as the denominator in Equation 6 contains κ*d_i_, but it could be that this is dissociable as u_j_ does not depend on κ (at least in some way as this implies d=0). Is there a way that the authors can come up with a paradigm, experiment or analyses to resolve this issue? If this is the case, this is an approach that could significantly advance the current state of the art in the field. The authors would need to convincingly demonstrate that this is the case.

3) The "toy" example based on the binomial model can be illustrative, it can also be confusing if some clarifications are not done from the beginning. From the moment the authors started to argue that LOTO can be used to detect trial to trial fluctuations on θ, it was clear to me that it was impossible to dissociate trial to trial fluctuations in *k* and θ. The explanation that this is the case comes only in the last paragraph of the subsection. I am not exactly sure what would be the best way to deal with this, but perhaps this issue about LOTO can be mentioned before the authors dive into applying LOTO to this example. Moreover, this is an issue that the authors might want to emphasize in the Discussion by explicitly describing what is or what is not possible with LOTO and giving some recommendations in this respect.

4) The usual approach when modelling trial-to-trial fluctuations for a given parameter is to assume some parametrization (e.g. uniform distribution for the starting evidence or Gaussian distribution for drift rate, or sometimes a Gamma distribution for precision parameters). While the correlations between LOTO recovered and true values is rather high in general, it would be interesting to see whether the distribution of the parameters recovered by LOTO match the shape of the generating distribution. If yes, great; if not, why not?

5) In subsection “An example of using LOTO for model-based fMRI”, the authors argue that the neuroimaging effects were "even stronger than those reported in the original publication". These statements have to be backed up via quantitative analyses. Given that the authors are performing their analysis based on the SPM framework, a direct way to test this is via the Bayesian model comparison module. In this way one could understand how much stronger a given activation pattern is better explained by one model or the other. Please provide the statistical maps of these analyses.

6) In the LOTO analyses for subsection “An example of using LOTO for model-based fMRI” (reported in Appendix D), σ could have been included as a potential source of trial-to-trial fluctuations. Why was this not the case? If σ does not change from trial to trial, then one should see this from the null distribution analyses on potential improvement of deviance. If this is the case, then how are the neuroimaging results affected?

---

## [Author Response]

Summary:The authors discuss a leave-one-out method to estimate trial-by-trial variability in parameters of computational models (LOTO). The rationale is that the deviation between a model parameter that is estimated using all data and a model parameter that is estimated using all data minus one data point, reflects the influence of that one data point. The authors demonstrate that LOTO is relatively easy to implement and faster with respect to previous approaches, and importantly it can be easily demonstrated via simulations whether the adopted approach is falsifiable.

We thank the reviewers for this overall very positive evaluation of our work. In the following, we address the issues raised by the reviewers point by point. Note that the new scripts which were required to run the revision analyses are uploaded on our OSF project (https://osf.io/du85f/).

On a general note, we re-organized the manuscript to better match the structure of *eLife* articles in the following way: the previous chapter I is the “Introduction” now; the previous chapters II–XI belong to the “Results section” which contains 11 chapters (Results sections I–XI; including a new chapter; see below); the previous Appendices A–D constitute the “Materials and methods section” now. There are no supplementary files (except for the figure supplements).

Essential revisions:1) An important issue with the current version of the paper is that in subsection “Comparison of LOTO with related approaches”, a comparison with the more complex, but presumably superior method of joint modeling (Turner et al.) is not performed. It would be good to see how much information about the single trial parameter estimates is really lost when not accounting for the covariance structure as Turner proposes.

We agree that a comparison of LOTO with the joint modeling approach (Turner et al., 2015) is desirable. In the revised manuscript, we added a comparison between LOTO and the joint modeling approach for estimating variability in parameter κ of the hyperbolic discounting (HD) model (we had problems implementing the joint modeling approach for the LBA model, which is not surprising given the high complexity of the approach).

Note that the joint modeling approach requires fitting the model via hierarchical Bayesian modeling (Farrell and Lewandowsky, 2018; Lee and Wagenmakers, 2013), because the behavioral and neural parameters of the joint model are assumed to be drawn from a (joint) higher-order (multivariate normal) distribution. Hence, implementing this approach allowed us to compare LOTO also with the (non-joint) Bayesian modeling approach itself, which was not included in the original manuscript either. Adding this method to the comparison also helped to quantify the contribution of accounting for the covariance structure more exactly.

The results are described at the end of subsection “Comparison of LOTO with related approaches” and are illustrated in Figure 9—figure supplement 1. Interestingly, at least for this example (i.e., parameter κ of the HD model) the correlation between the true trial-by-trial parameter values and LOTO’s estimates are slightly higher than those for the Bayesian modeling and the joint modeling approaches, while the joint modeling approach outperforms the Bayesian approach. On the other hand, the correlation between the neural signal and LOTO’s estimates is slightly lower than the correlation of the joint modeling approach. Again, the joint modeling approach outperforms the non-joint Bayesian approach with respect to the neural data. LOTO and the non-joint Bayesian approach yield similar results. Overall, we deem the differences between the three approaches rather small (for our simulations with 30 participants and 160 trials).

2) A substantial concern about the utility of the approach and the interpretation of the results obtained by LOTO comes from the fact that recent evidence indicates that a good proportion of trial to trial fluctuations comes from inference processes of the incoming signals or subsequent computations (for emerging evidence see e.g., Drugowitsch et al., 2016; Findling et al., 2018; Polania et al., 2018). This suggests, that for instance, in the case of the HD example given by the authors, fluctuations can come directly from encoding d_i_ or x_i_ or even from performing the computation (x_i_ /(1+κ*d_i_)) but not necessarily from κ. This means that variability in behavior can be misattributed to a given parameter (and therefore misattributed to a brain region when applied to neural data), but the sources of fluctuations may potentially come from encoding/decoding processes of input-signals/computations as indicated by recent research. Is there a way that the authors can rule out or study this possibility? For the case of the HD model, I assume that this might be difficult as the denominator in Equation 6 contains κ*d_i_, but it could be that this is dissociable as u_j_ does not depend on κ (at least in some way as this implies d=0). Is there a way that the authors can come up with a paradigm, experiment or analyses to resolve this issue? If this is the case, this is an approach that could significantly advance the current state of the art in the field. The authors would need to convincingly demonstrate that this is the case.

Indeed, dissociating trial-by-trial variability of input or output signals of computational processes from variability in parameters of these processes is a very interesting potential application of LOTO. To see whether LOTO could be useful in this regard, we tested whether inducing variability in the input variables of the HD model (i.e., the amount *x* and delay *d* of the delayed option) or in its output variable (i.e., the utility *u*) would be misattributed to parameter κ, when applying LOTO. Mostly in line with the reviewer’s intuitions, this worked well for the amount *x*, but for high levels of variability in delay *d* or in utility *u*, a potential misattribution could not be ruled out (see Figure 5—figure supplement 1).

Next, we reasoned that a better dissociation between the parameter and the input/output signals might be achieved when adding more choice options. The intuition is that whereas parameter κ could be assumed to be stable across different options (and to vary only from trial to trial), variability in the input/output variables could be assumed to vary across options. Hence, variability in the input/output variables should be less likely to be taken up by κ to vary from trial to trial via LOTO.

Therefore, we set up simulations of an adapted paradigm of the intertemporal choice task with three options (see also Gluth et al., 2017) and applied the “non-parametric test for systematic variability” for the cases of true variability in κ, *x, d*, or *u* (for details, see subsection “An example of using LOTO for model-based fMRI”). As predicted, a significant improvement in model fit over and above the improvement under the null hypothesis of no variability was only found for κ, but not for any of the input/output variables (see Figure 5—figure supplement 2).

Taken together, LOTO might indeed be a powerful tool to dissociate parameter variability from variability of input and output signals of a neurocognitive process. As with many other potential applications of LOTO, however, this requires appropriate experimental designs and careful simulations. We discuss these issues at the end of subsection “A non-parametric test for the presence of systematic trial-by-trial variability”.

3) The "toy" example based on the binomial model can be illustrative, it can also be confusing if some clarifications are not done from the beginning. From the moment the authors started to argue that LOTO can be used to detect trial to trial fluctuations on θ, it was clear to me that it was impossible to dissociate trial to trial fluctuations in k and θ. The explanation that this is the case comes only in the last paragraph of the subsection. I am not exactly sure what would be the best way to deal with this, but perhaps this issue about LOTO can be mentioned before the authors dive into applying LOTO to this example. Moreover, this is an issue that the authors might want to emphasize in the Discussion by explicitly describing what is or what is not possible with LOTO and giving some recommendations in this respect.

We followed the reviewer’s suggestion and now state up front that one reason why we dub the binomial distribution a “toy” model is because LOTO cannot infer more information about parameter θ than what is given by the observations *k* (subsection “A “toy” model example”). Our recommendation in this regard, which we re-iterate in the “LOTO recipe” (subsection “The LOTO recipe”), is to include the behavioral data themselves (i.e., choices, RT) when running simulations and regressing the true parameter values onto LOTO estimates (as we have done in our own analyses; see Figure 6C and Figure 12A). If LOTO does not provide any additional information beyond the observations themselves, this will become evident in these regression analyses.

4) The usual approach when modelling trial-to-trial fluctuations for a given parameter is to assume some parametrization (e.g. uniform distribution for the starting evidence or Gaussian distribution for drift rate, or sometimes a Gamma distribution for precision parameters). While the correlations between LOTO recovered and true values is rather high in general, it would be interesting to see whether the distribution of the parameters recovered by LOTO match the shape of the generating distribution. If yes, great; if not, why not?

This is indeed an interesting question, and we denoted a new subsection “Comparison of LOTO with related approaches” to this topic. In one sentence: LOTO does not recover the distribution of the underlying parameters.

This becomes most apparent when going back to the initial “toy” model of the binomial distribution. For this model, we sampled 500 blocks of 20 trials each with probability parameter θ being fixed over each 20 trials but varying across the 500 blocks (similar to what we did in subsection “A “toy” model example”). Across these 500 blocks, we once sampled values of θ from a uniform distribution and once from a normal distribution. Independent of the underlying distribution, the distribution of LOTO estimates resembled a binomial distribution (see the new Figure 8). This is because the LOTO estimates are perfectly correlated with the observations *k* (see subsection “A “toy” model example”), which are binomially distributed by definition.

Stated differently, the parameter and LOTO are linked via the model, and the likelihood function of the model determines whether and how much a specific parameter value exerts an influence on the observations (and thus an influence on LOTO’s estimates). Hence, the distribution of LOTO’s estimates do not depend so much on the distributions of the parameter values but rather on the “error function” of the model (e.g., the binomial distribution, the soft-max function for a choice model, the Wiener distribution of a diffusion model, etc.).

Because the “toy” model of the binomial distribution is neither particularly interesting nor suitable for LOTO-based inferences (see our reply to the previous point), we also looked at the distribution of LOTO values for the two parameters κ (i.e., discount factor) and *b* (i.e., decision threshold) of the HD-LBA model (i.e., the combination of hyperbolic discounting with the LBA). Again, we drew single-trial parameter values of κ and *b* either from normal or from uniform distributions. Here, the LOTO values are distributed (more or less) normally (see Figure 8—figure supplements 1 and 2, for κ and *b*, respectively). Again, the reason for this is that the parameter values are connected with LOTO and the observations (i.e., choices and RT) via the model which exhibits a roughly normally distributed error term (due to the drift rates which are drawn from normal distributions). Importantly, the shape of the underlying distribution did not affect the strength of the correlation between true parameters and LOTO’s estimates much (see the lowest panels in Figure 8—figure supplements 1 and 2). Thus, we conclude that on the one hand LOTO is not suitable to infer the underlying distribution from which fluctuating parameter values are possibly drawn. On the other hand, however, cognitive neuroscientists who are primarily interested in using LOTO to detect significant relationships between cognitive models and neural data (i.e., the audience we have in mind) do not need to be concerned (much) about these underlying distributions.

5) In subsection “An example of using LOTO for model-based fMRI”, the authors argue that the neuroimaging effects were "even stronger than those reported in the original publication". These statements have to be backed up via quantitative analyses. Given that the authors are performing their analysis based on the SPM framework, a direct way to test this is via the Bayesian model comparison module. In this way one could understand how much stronger a given activation pattern is better explained by one model or the other. Please provide the statistical maps of these analyses.

We agree that such a statement would require statistical confirmation. However, we decided to refrain from performing the proposed Bayesian model comparison, because the comparability of the current analysis and that of the original publication is limited anyway. This is because the analyses do not only differ with respect to the method of capturing trial-by-trial variability (LOTO vs. Gluth and Rieskamp, 2017), but also with respect to the way the model is fitted in the first place (maximum likelihood vs. Bayesian modeling) and with respect to the fMRI software package (SPM8 vs. SPM12). Hence, a potential significant difference between the analyses could not be exclusively attributed to the different methods of capturing trial-by-trial variability.

Therefore, we decided to take out the claim that the current results are stronger than those of the original publication. Instead, we only mention that the results are mostly in line with the original study, in which a relationship between trial-by-trial memory and hippocampal activation has been found as well subsection “An example of using LOTO for model-based fMRI”.

6) In the LOTO analyses for subsection “An example of using LOTO for model-based fMRI” (reported in Appendix D), σ could have been included as a potential source of trial-to-trial fluctuations. Why was this not the case? If σ does not change from trial to trial, then one should see this from the null distribution analyses on potential improvement of deviance. If this is the case, then how are the neuroimaging results affected?

We did not include parameter σ as a source of trial-by-trial variability for theoretical reasons: note that in this example (memory-based decisions), we do not really track trial-by-trial but rather item-by-item variability. Each snack item is presented three times in the decision-making task, and we assume the same ability to remember the snack across the three trials (which is sensible, given that it is always the same snack). On the other hand, parameter σ can be seen as quantifying “noise in the decision process”, and it seems to be more plausible to assume that this noise differs in every trial.

Despite these conceptual objections, we decided to test whether true trial-by-trial variability in σ would be misattributed to the memory-related parameters (α and β) when applying LOTO. The results suggest that such a misattribution is unlikely to have occurred: The LOTO-based improvement of model fit in the presence of true variability in σ does not exceed the null distribution and there is also no systematic relationship between the amount of variability in σ and the model fit. We report the results in subsection “An example of using LOTO for model-based fMRI” and Figure 12—figure supplement 1.